# *Schistosoma mansoni* α-N-acetylgalactosaminidase (SmNAGAL) regulates coordinated parasite movement and egg production

**Benjamin J. Hulme**[1], **Kathrin K. Geyer**[1], **Josephine E. Forde-Thomas**[1], **Gilda Padalino**[1], **Dylan W. Phillips**[1], **Wannaporn Ittiprasert**[2], **Shannon E. Karinshak**[2], **Victoria H. Mann**[2], **Iain W. Chalmers**[1], **Paul J. Brindley**[2], **Cornelis H. Hokke**[3], **Karl F. Hoffmann**[1] *

**1** Institute of Biological, Environmental and Rural Sciences (IBERS), Edward Llwyd Building, Aberystwyth University, Aberystwyth, United Kingdom, **2** Department of Microbiology, Immunology and Tropical Medicine, Research Center for Neglected Diseases of Poverty, School of Medicine and Health Sciences, George Washington University, Washington DC, United States of America, **3** Department of Parasitology, Leiden University Center for Infectious Diseases, Leiden University Medical Center, Leiden, The Netherlands

* krh@aber.ac.uk

**Data Availability Statement:** The full coding sequence of Smp_089290 is deposited under the GenBank accession number MZ508282. Raw

## Abstract

α-galactosidase (α-GAL) and α-N-acetylgalactosaminidase (α-NAGAL) are two glycosyl hydrolases responsible for maintaining cellular homeostasis by regulating glycan substrates on proteins and lipids. Mutations in the human genes encoding either enzyme lead to neurological and neuromuscular impairments seen in both Fabry- and Schindler/Kanzaki- diseases. Here, we investigate whether the parasitic blood fluke *Schistosoma mansoni*, responsible for the neglected tropical disease schistosomiasis, also contains functionally important α-GAL and α-NAGAL proteins. As infection, parasite maturation and host interactions are all governed by carefully-regulated glycosylation processes, inhibiting *S. mansoni*'s α-GAL and α-NAGAL activities could lead to the development of novel chemotherapeutics. Sequence and phylogenetic analyses of putative α-GAL/α-NAGAL protein types showed Smp_089290 to be the only *S. mansoni* protein to contain the functional amino acid residues necessary for α-GAL/α-NAGAL substrate cleavage. Both α-GAL and α-NAGAL enzymatic activities were higher in females compared to males ($p<0.05$; α-NAGAL > α-GAL), which was consistent with *smp_089290*'s female biased expression. Spatial localisation of *smp_089290* revealed accumulation in parenchymal cells, neuronal cells, and the vitellaria and mature vitellocytes of the adult schistosome. siRNA-mediated knockdown (>90%) of *smp_089290* in adult worms significantly inhibited α-NAGAL activity when compared to control worms (*siLuc* treated males, $p<0.01$; *siLuc* treated females, $p<0.05$). No significant reductions in α-GAL activities were observed in the same extracts. Despite this, decreases in α-NAGAL activities correlated with a significant inhibition in adult worm motility as well as in egg production. Programmed CRISPR/Cas9 editing of *smp_089290* in adult worms confirmed the egg reduction phenotype. Based on these results, Smp_089290 was determined to act predominantly as an α-NAGAL (hereafter termed SmNAGAL) in

sequence files from the NGS libraries are available at the Sequence Read Archive (SRA) under BioProject ID PRJNA743897.

**Funding:** This research was funded in whole, or in part, by the Wellcome Trust (https://wellcome.org; Grant number 107475/Z/15/Z) to KFH. At George Washington University Medical School, schistosome-infected mice and snails were provided by the NIAID Schistosomiasis Resource Center of the Biomedical Research Institute, Rockville, Maryland through NIH-NIAID Contract HHSN272201000005I for distribution through BEI Resources. BJH was supported from a kind donation provided by David & Eleanor James. The funders had no role in study design, data collection and analysis, decision to publish, or preparation of the manuscript.

**Competing interests:** The authors have declared that no competing interests exist.

schistosome parasites where it participates in coordinating movement and oviposition processes. Further characterisation of SmNAGAL and other functionally important glycosyl hydrolases may lead to the development of a novel anthelmintic class of compounds.

## Author summary

Schistosomiasis is a parasitic disease caused by infection with blood flukes, which leads to acute and chronic pathology in millions of infected individuals located in deprived tropical and subtropical regions. Elucidating the function of schistosome genes has provided a clearer view on their roles in various molecular pathways, which are critical to successful parasitism. This information is invaluable when progressing novel drug and vaccine candidates. Here, we add to the existing knowledge of the *Schistosoma mansoni* parasitic glycan processing and modification machinery by functionally characterising a glycosyl hydrolase (*S. mansoni* α-N-acetylgalactosaminidase, SmNAGAL). We demonstrate that this protein is enzymatically active and important in coordinating parasite movement in adult male and female schistosomes. Additionally, we provide evidence that this protein regulates pathways associated with egg production in female schistosomes, which is responsible for inducing pathological reactions. Developing drugs that inhibit SmNAGAL enzymatic activity could provide a novel approach for controlling schistosomiasis.

## Introduction

Schistosomiasis is a parasitic disease caused by infection with blood flukes of the genus *Schistosoma* and is estimated to affect more than 250 million people worldwide [1]. With up to 200,000 deaths per annum, schistosomiasis is considered one of the most devastating neglected tropical diseases (NTDs) of resource poor communities within developing countries [2]. Currently, the primary method used to control schistosomiasis is mediated by praziquantel chemotherapy [3] with recent data suggesting that this drug targets transient receptor potential (TRP) channels expressed in adult schistosomes [4,5]. The continued use of praziquantel in mass drug administration (MDA) programmes has given rise to the fear of possible drug resistant blood flukes developing. Indeed, multiple studies have demonstrated that praziquantel insensitive schistosomes can be generated in the laboratory [6–8]. These findings, thus, highlight the need for the development of novel anti-schistosomal drugs, which can act as praziquantel replacements if praziquantel-resistant schistosomes eventually arise. Identifying and inhibiting processes critical to schistosome developmental biology and/or host interactions presents a strategy that may contribute to this objective.

It is well established that schistosomes carefully regulate their production of specific glycans and glycan elements as they transition between diverse environmental niches including water, mammals and snails [9–12]. Some parasite glycans, as moieties of glycoproteins or glycolipids, are immunomodulatory in nature and contribute to host (snail/mammal) as well as dioecious (male/female schistosome) interactions [12–15]. Regardless of their significance in orchestrating these heterospecific, conspecific or developmental activities, the gene products responsible for schistosome glycoprotein and glycolipid anabolism and degradation are only slowly being characterised [16,17]. One such glycogene family, critical to glycosylation homeostasis in other eukaryotes, is the glycosyl hydrolases (or glycoside hydrolases, GH). These enzymes hydrolyse glycosidic bonds between monosaccharide residues in a carbohydrate chain or bonds between

the carbohydrate and a non-carbohydrate moiety (such as a lipid or protein). According to the latest survey of glycogene diversity (Carbohydrate-Active enZYmes, CAZY, database [18]), the GHs are comprised of approximately 171 families (further segregated into clans [19]), each differing in substrate specificity and molecular mechanism.

GH family 27 comprises mainly α-galactosidases (α-GAL) and α-N-acetylgalactosaminidases (α-NAGAL) responsible for the cleavage of terminal α-D-galactose and α-N-acetylgalactosamine residues from glycosylated substrates, respectively [20,21]. Molecular defects in *Homo sapiens α-gal* lead to a lysosomal storage condition called Fabry disease, which is characterised by a range of multi-organ pathologies due to an accumulation of galactose-containing substrates (mostly aberrant glycolipids) in tissues [22]. Mutations in *H. sapiens α-nagal* lead to a clinically heterogeneous lysosomal disorder called Schindler/Kanzaki disease, which can contribute to neuromuscular defects and skin disorders due to accumulation of both glycolipids and glycoproteins [23,24]. While these data strongly support the indispensable nature of both α-GAL and α-NAGAL activities in *H. sapiens*, RNA interference (RNAi)–mediated knockdown studies of *gana-1* (an ortholog of both *H. sapiens α-gal* and *α-nagal*) in *Caenorhabditis elegans* failed to demonstrate essentiality [25]. As the significance in loss of α-GAL and α-NAGAL activities may phenotypically vary among species, we initiated an investigation exploring the identification and characterisation of *S. mansoni* genes with similarity to these two members of GH family 27.

Of the five putative *S. mansoni* α-GAL/α-NAGAL proteins identified, we present sequence-, bioinformatic, functional genomic- and enzymatic- based evidence suggesting that Smp_089290 is the only α-GAL/α-NAGAL ortholog present in the genome (v 7.0) of this blood fluke. In the adult schistosome, *smp_089290* transcript abundance and enzymatic activity (α-NAGAL > α-GAL) were both enriched in the female compared to the male. Spatial localisation of *smp_089290* by whole-mount *in situ* hybridisation demonstrated accumulation to the vitellaria and vitellocytes in adult female schistosomes as well as diffuse distribution throughout parenchymal cells in both adult male and female parasites. Furthermore, RNAi-mediated knockdown of *smp_089290* led to a significant reduction in adult worm motility and egg production (confirmed by CRISPR/Cas9 genome editing). Collectively, our results predict that Smp_089290/SmNAGAL is a predominant α-NAGAL necessary for coordinated schistosome movement and oviposition processes.

## Materials and methods

### Ethics statement

All mouse procedures performed at Aberystwyth University (AU) adhered to the United Kingdom Home Office Animals (Scientific Procedures) Act of 1986 (project licenses PPL 40/3700 and P3B8C46FD) as well as the European Union Animals Directive 2010/63/EU, and were approved by AU's Animal Welfare and Ethical Review Body (AWERB). The use of mice at George Washington University Medical School was approved by The Institutional Animal Care and Use Committee (IACUC).

### Parasite material

A Puerto Rican strain of *S. mansoni* (NMRI, Naval Medical Research Institute) was used in this study. Mixed-sex adult worms were perfused from percutaneously infected TO (HsdOla: TO, Envigo, UK), MF-1 (HsdOla:MF1, Envigo, UK) or Swiss Webster (CFW, Charles River Laboratories, USA) mice infected 7 wk earlier with 180 cercariae [26]. These parasites were cultured (10 sex-separated adults or five adult pairs per well in a 48 well tissue culture plate) at 37˚C in DMEM (Sigma-Aldrich) supplemented with 10% foetal calf serum, 2 mM L-glutamine

and 100 μg/ml penicillin/streptomycin in an atmosphere of 5% $CO_2$ with a 70% media exchange performed after 24 hr and 48 hr for adult male/adult pairs and female worms respectively. Cultured worms were subsequently used for RNA interference, CRISPR/Cas9 genome editing, total RNA isolation/complementary DNA (cDNA) generation, dual RNA and genomic DNA (gDNA) extraction and enzyme assays.

## Multiple sequence alignment and phylogenetic trees

The Pfam identifier PF16499 (melibiase-2) was used to identify *S. mansoni*, *Schistosoma haematobium* and *Schistosoma japonicum* GH family 27 orthologs of *H. sapiens* α-GAL and α-NAGAL enzymes using WormBase ParaSite [27]. Only those *Schistosoma* entries that contained >70% of the residues within the melibiase-2 domain identified by Pfam, UniProt and CAZypedia were retained [18,28,29]. These included *S. mansoni* Smp_170840 (G4VLE1), *S. haematobium* MS3_10002 (A0A095A3Q8), *S. haematobium* MS3_10345 (A0A095B3P1), *S. mansoni* Smp_179250 (G4VLD7), *S. mansoni* Smp_247760 (A0A5K4F2K9), *S. mansoni* Smp_247750 (A0A5K4F165), *S. haematobium* MS3_10001 (A0A095CF43), *S. japonicum* EWB00_005283 (A0A4Z2D326), *S. japonicum* EWB00_005285 (A0A4Z2D213), *S. japonicum* EWB00_005284 (A0A4Z2D211), *S. haematobium* MS3_11280 (A0A095BSM8) and *S. mansoni* Smp_089290 (G4VLE3). A multiple sequence alignment of these *Schistosoma* melibiase-2 domain containing proteins with human α-GAL (*H. sapiens*, Hs_GAL, P06280), human α-NAGAL (*H. sapiens*, Hs_NAGAL, P17050), chicken α-NAGAL (*Gallus gallus*, Gg_NAGAL, Q90744) and *C. elegans* α-NAGAL (Ce_GANA-1, Q21801) was generated using MUSCLE v3.8 [30]. Amino acids included in the alignment were based on the positions of ligand binding and catalytic aspartic acid (D) residues of the melibiase-2 domains of human α-GAL and α-NAGAL [31–33].

Phylogenetic analysis of melibiase-2 domain containing proteins from a broad range of species across phyla was conducted using Bayesian inference and Maximum Likelihood approaches. MUSCLE v3.8 software was used to align proteins from *S. mansoni*, *S. haematobium*, *S. japonicum* (all noted previously) and additional sequences including *H. sapiens* Hs_GAL (P06280), *H. sapiens* Hs_NAGAL (P17050), *Drosophila melanogaster* Dm_CG7997 (Q7K127), *D. melanogaster* Dm_CG5731 (Q8MYY3), *C. elegans* Ce_GANA-1 (Q21801), *Arabidopsis thaliana* At_AGAL1 (Q9FT97), *A. thaliana* At_AGAL2 (Q8RX86), *A. thaliana* At_AGAL3 (Q8VXZ7), *Schizosaccharomyces pombe* Sp_MEL1 (Q9URZ0), *Coffea arabica* Ca_GAL (Q42656), *G. gallus* Gg_NAGAL (Q90744), *Rattus norvegicus* Rn_NAGAL (Q66H12), *Mus musculus* Mm_GAL (P51569) and *M. musculus* Mm_NAGAL (Q9QWR8). The alignment was submitted to Gblocks Server 0.91b which removed non-conserved regions to allow phylogenetic comparisons [34]. In total, 208 amino acid residues were used for this phylogenetic analysis. Bayesian inference analyses were performed using MrBayes v3.2.6 [35], the WAG substitution model [36], 500,000 generations and a sample frequency of 100. As the analysis proceeded, log likelihood values were generated and plotted against the number of generations. The analysis was stopped upon examination of the average standard deviation of split frequencies indicating when the log likelihood values reached a stationary distribution. Maximum Likelihood analyses were performed using MEGA X [37] with the JTT model [38] and 500 bootstrap replicates. The final Bayesian consensus phylogram was generated using FigTree v1.4.3 [39]. The Bayesian inference tree was edited using Adobe Illustrator CS4 software where Bayesian posterior probability support values (outside parentheses) and Maximum Likelihood percentage bootstrap support values (inside parentheses) were superimposed on corresponding nodes.

## Homology modelling of Smp_089290

The three-dimensional structure of Smp_089290 was derived by homology modelling using MODELLER [40]. The α-GAL/α-NAGAL template selected for Smp_089290 modelling was the three-dimensional structure of *H. sapiens* α-NAGAL (Protein Databank identification code 4DO4) [32]. The sequence identity between Smp_089290 and the human α-NAGAL was 40% with a sequence coverage of 97%, hence well within the acceptable range for comparative modelling [41]. The stereochemical quality of the Smp_089290 model was assessed by RAMPAGE (Ramachandran Plot Analysis) [42], ProSA-web (Protein Structure Analysis) [43] and Verify 3D [44].

## Total RNA isolation and cDNA generation

*S. mansoni* total RNA was isolated using the Direct-zol RNA MiniPrep Kit (Zymo Research) with slight modifications. Briefly, worms were removed from culture, transferred to 2 ml Eppendorf tubes, and homogenised in QIAZOL reagent (Qiagen) with a 5 mm diameter stainless steel bead (Qiagen) for a total of 6 min (2 x 3 min with 1 min on ice in between homogenisations) at 50 Hz using a TissueLyser LT (Qiagen). Thereafter, the RNA was treated with DNase I to remove contaminating gDNA. RNA was eluted into collection tubes by adding 30 μl of DNase/RNase-free H$_2$O directly to the Direct-zol column matrix and subsequently centrifuged at 17,000 x *g* for 60 sec. Yields of total RNA samples were assessed using a Nano-Drop ND-1000 UV-Vis spectrophotometer; RNAs were concentrated when necessary in a Concentrator plus (Eppendorf) and re-quantified. Schistosome cDNAs were reverse transcribed from the total RNAs using the SensiFAST cDNA Synthesis Kit (Bioline) according to the manufacturer's instructions.

## cDNA cloning and sequencing of *smp_089290*

PCR primers used to amplify the full coding sequence of Smp_089290 included forward (5′–ATG GCT ACC GTA CCA CCG– 3′) and reverse (5′–CTA TAA CAA TGT CTG AAA CAG TCC ATC– 3′) pairs. PCR amplification utilised cDNA prepared from total RNA obtained from mixed-sex adult worms [45]. The amplification was performed in 25 μl containing Biomix reaction mix (Bioline) and ultra-stable Taq DNA polymerase (Bioline). Amplicons were subjected to electrophoresis through 1% w/v agarose, stained and visualised with SYBR Safe dye (Thermo Fisher Scientific) under UV light. Products of the predicted size of 1,463 bp were ligated into pGEM-T Easy (Promega) after which α-Select Bronze Efficiency Competent Cells (Bioline) were transformed with the ligation products. Blue/white screening of *E. coli* colonies was accomplished on LB agar plates supplemented with 5-bromo-4-chloro-3-indolyl-β-D-galactopyranoside (X-Gal) and Isopropyl β-D-1-thiogalactopyranoside (IPTG). Recombinant (white) clones were isolated from a 5 ml High Salt (HS) Luria Bertani (LB) culture containing ampicillin (50 μg/ml) using the QIAprep Spin Miniprep Kit (Qiagen). Inserts in pGEM-T Easy were sequenced at Aberystwyth University's Translational Genomics Facility and the sequence trace was analysed using FinchTV 1.4.0 software [46]. The full coding sequence of Smp_089290 is deposited under the GenBank accession number MZ508282.

## DNA microarray analysis

The *S. mansoni* long oligonucleotide DNA microarray was designed and constructed by Fitzpatrick and colleagues [45]. The DNA microarray consists of 35,437 *S. mansoni* oligonucleotide 50-mers as well as 2,195 controls and is deposited in the ArrayExpress database under the accession number A-MEXP-830. *S. mansoni* lifecycle stages profiled by the DNA microarray

included egg, miracidia, mother (2 day) sporocysts, daughter sporocysts, cercariae, 3 hr schistosomula, 24 hr schistosomula, 3 day schistosomula, 6 day schistosomula, 3 wk worms, 5 wk worms, 7 wk worms, male 7 wk worms and female 7 wk worms. Normalised fluorescence intensity values (available via ArrayExpress under the experimental accession number E-MEXP-2094), corresponding to 50-mers representing exons 2 and 3 of *smp_089290*, enabled the quantification of gene expression profiles across these developmental stages of the schistosome.

### RNA-Seq meta data analysis

The RNA-Seq meta database was created by Lu and colleagues [47] by normalising gene expression values derived from RNA-Seq data produced by various publications. Gene expression values for each lifecycle stage were obtained from the following reports: Anderson *et al.* [48] for egg, Wang *et al.* [49] for miracidia and sporocysts, Protasio *et al.* [50] for cercariae, 3 hr and 24 hr schistosomula, Protasio *et al.* [51] for 21 day juvenile male, 21 day juvenile female, 28 day juvenile male, 28 day juvenile female, 35 day adult male, 35 day adult female, 38 day adult male and 38 day adult female and Lu *et al.* [52] for 42 day adult male and 42 day adult female. The normalised gene expression values for a gene of interest were obtained by entering the gene ID into the 'schisto_xyz' search engine [53] and subsequently plotted as a gene expression profile.

### RNA interference (RNAi)

Small interfering RNA (siRNA) duplexes were designed (si*Smp_089290*: sense strand 5'-CUA AUG AAA UCG UUG CAG A-3'; anti-sense strand 5'-UCU GCA ACG AUU UCA UUA G-3') based on the cDNA sequence verified *smp_089290* amplicon and possessed no significant homology to other gene products (i.e. less than 15 out of 19 nucleotides conserved) in the *S. mansoni* genome assembly (v7.0) [54,55]. An siRNA duplex designed for firefly luciferase (si*Luc*) without significant homology to gene products in the *S. mansoni* genome assembly (v7.0) served as a negative control [56]. Briefly, sets of 10 worms (sex-separated) or five adult pairs were transferred to 0.4 cm pathway electroporation cuvettes (Invitrogen) containing DMEM supplemented with 2 mM L-glutamine and 100 μg/ml penicillin/streptomycin. siRNA duplexes (5 μg) were subsequently added and worms were electroporated with a single pulse at 125 V (LV mode) for 20 ms using an ECM 830 Square Wave Electroporation System (BTX, Harvard Apparatus, Holliston, MA). Worms were subsequently transferred to a 48 well tissue culture plate and cultured for up to seven days.

### Preparation of CRISPR/Cas9 plasmid constructs

Two CRISPR target sites (single guide RNA; sgRNA sequence) for Cas9-catalysed gene editing for *smp_089290* were designed by Breaking-Cas [57] with default parameters compatible for the protospacer adjacent motif (PAM) of Cas9 from *Streptococcus pyogenes* (NGG) [58, 59]. Two sgRNAs targeting the coding regions exons 1 (SmNAGALX1: 5′–CUA CCG UAC CAC CGA UGG GU– 3′) and 2 (SmNAGALX2: 5′–UUG UAA UCU AUG GCG UAU GC– 3′) were used in this study. The sgRNAs contained >40% GC-content, no self-complementarity and no off-target sites against the *S. mansoni* genome assembly (v7.0) as predicted by Breaking-Cas software. A 20 nucleotide (nt) 'Scramble' sgRNA designed with low homology to the *S. mansoni* genome assembly (v7.0) and lack of an adjacent PAM site (necessary for Cas9 function) served as a non-targeting control (5′–GCA CUA CCA GAG CUA ACU CA– 3′). CRISPR/Cas9 plasmid constructs were assembled using the pLenti-Cas-Guide construction Kit (GE100010, OriGene, Maryland, USA) and each sgRNA was ligated into the pLenti-Cas-

sgRNA backbone as per the manufacturer's instructions. Expression of the sgRNA (SmNA-GALX1, SmNAGALX2 or Scramble containing plasmids) was driven by the mammalian U6 promoter and expression of Cas9 from *S. pyogenes* with nuclear localisation signals was driven by the human cytomegalovirus (CMV) promoter. Each CRISPR/Cas9 plasmid construct was independently transformed into DH5-α Chemically Competent Cells (GoldBio) with chloramphenicol (25 µg/ml) as drug selection. The chloramphenicol resistant transformants were confirmed for corrected orientation of sgRNA in the vector backbone by Sanger direct sequencing. To amplify the plasmid DNA, the transformants were first cultured in a 10 ml HSLB culture containing chloramphenicol (25 µg/ml) in a shaking incubator at 37°C, 225 rpm for approximately 7 hr. Thereafter, 10 ml of the culture was added to 240 ml of fresh HSLB culture containing chloramphenicol (25 µg/ml) and cultured overnight, as above. Plasmid DNA was recovered from 150 ml of this overnight culture using the GenElute HP Plasmid Midiprep Kit (Sigma-Aldrich) as per the manufacturer's instructions except for a modified elution step; here, accomplished with 800 µl of nuclease-free $H_2O$. Plasmid DNAs were quantified using the NanoDrop ND-1000 UV-Vis spectrophotometer (Thermo-Fisher Scientific), after which the DNA was precipitated and subsequently dissolved in Opti-MEM Reduced Serum Medium (Thermo-Fisher Scientific) and stored at -20°C.

## CRISPR/Cas9 mediated genome editing

A mixture of 24 µg of CRISPR/Cas9 plasmid DNA (either SmNAGALX1, SmNAGALX2 or Scramble) reconstituted in a total volume of 400 µl Opti-MEM Reduced Serum Medium was dispensed into a 0.4 cm pathway electroporation cuvette (Invitrogen). Five pairs of worms were transferred to the cuvettes, which had been chilled on wet ice, and subjected to square wave electroporation with a single pulse of 125 V (LV mode) for 20 ms (BTX, Harvard Apparatus). Worms were subsequently transferred to a 48 well tissue culture plate and cultured for up to seven days at 37°C in an atmosphere of 5% $CO_2$. Upon completion of the seven day experiment, RNA and gDNA was extracted from adult male and female worms using both RNAzol reagent (Sigma-Aldrich) and DNAzol reagent (Invitrogen) based on the manufacturer's instructions. One ml of RNAzol reagent was added to the parasite material, then subsequently homogenised four times with a 5 mm diameter stainless steel bead (Qiagen) for 3 min at 50 Hz using a TissueLyser LT (Qiagen) with 1 min incubation on ice in between homogenisations. Once the tissues of the schistosomes were completely disrupted, the protocol steps were followed until completion. RNA samples were immediately used for cDNA synthesis, as above. gDNA samples were amplified by PCR using primers encompassing the programmed double-strand break (DSB) site at exon 1 (Forward: 5´–CTT ATA GGT GTG CCA TAT TAA CGA T– 3´, Reverse: 5´–ATG CAC TAC ATT CGA AAG ACA– 3´) or exon 2 (Forward: 5´–AGT GTT CTC ATG CAG TTA TCC T– 3´, Reverse: 5´–TCC ATG TCA GCT GAG ATC A– 3´) of *smnagal*. Correct amplification was verified by 1% w/v agarose gel electrophoresis. Thereafter, amplicons were subjected to the QIAseq 1-Step Amplicon Library Kit (Qiagen) for Illumina compatible next generation sequencing (NGS) library construction with GeneRead DNAseq Targeted Panels V2 (Qiagen) as per the manufacturer's instructions. Amplicon size of each NGS library was verified using a 2100 Bioanalyzer (Agilent, Santa Clara, CA). The NGS libraries were quantified using the GeneRead Library Quant Kit (Qiagen). NGS libraries were pooled at the Genewiz NGS facility (Genewiz, NJ) and processed with a MiSEQ configuration of 2x250 bp. The demultiplexing data generated by the Genewiz NGS facility was exported as Fastq files (.*qz* format). The mutations around the cut sites were analysed by CRISPResso2 software using default parameters [60,61]. Background mutations (i.e. mutations not attributable to genome editing) were inferred by identifying mutations present in the control

samples and the treatment samples. The unique mutations around the programmed DSB sites only reported in the high quality sequence reads of the target sample were designated as CRISPR/Cas9-induced mutations. The software analysed and reported the percentage of indels (insertions and deletions) and substitutions in the target sample (inferred by the number of reads that are modified in comparison to the total reads), the percentages of each mutation type and a list of each specific mutation showing which nts have been altered. Sequence reads from the NGS libraries are available at the Sequence Read Archive (SRA) under BioProject ID PRJNA743897.

## Quantitative reverse transcription (RT)–PCR (qRT-PCR) analysis

cDNAs synthesised from freshly perfused, siRNA treated or CRISPR/Cas9 plasmid treated schistosomes were used as templates for qRT-PCR to analyse transcript abundance. *smp_089290* transcript levels in freshly perfused schistosomes were quantified relative to Ras-Related GTP-binding protein D (*smrgbd*) and Tubulin-specific chaperone D (*smtscd*); these two genes were identified by Normfinder [62] as being constitutively expressed across nine *S. mansoni* lifecycle stages, including the adult [63]. *smp_089290* transcript levels in siRNA treated and CRISPR/Cas9 plasmid treated schistosomes were quantified relative to α-tubulin (*smat1*). qRT-PCR was performed using a StepOnePlus Real-Time PCR System (Applied Biosystems) and SensiFAST SYBR Hi-ROX mix (Bioline). Total reaction volume was 10 µl with 150 nM of each primer, 5 µl of SensiFAST SYBR Hi-ROX mix, 2 µl of cDNA template and 2.7 µl of nuclease-free $H_2O$. qRT-PCR primers for *smp_089290* were designed based on the sequence verified *smp_089290* amplicon and included forward (5'-CAC GAC TGA TGG TGG TGG-3') and reverse (5'-CTC GAT ACA TCA TTA TCC CGC T-3') pairs. *smrgbd* primers included forward (5'-CGG CTT TAA CTC GCC TAC AC-3') and reverse (5'-CAT TCG ACG GTT GTT CAC AC-3') pairs. *smtscd* primers included forward (5'-GCC AAC AAA TTT CGT GGT CT-3') and reverse (5'-TTC ACC ATT TCG GTC GTA CA-3') pairs. *smat1* primers included forward (5'-CTT CGA ACC AGC AAA TCA GA-3') and reverse (5'-GAC ACC AAT CCA CAA ACT GG-3') pairs. To determine *smp_089290* knockdown (KD) in adult worms following RNAi and CRISPR/Cas9 programmed knockout, *smp_089290* transcript levels in siRNA treated samples (at 48 hr post electroporation) and CRISPR/Cas9 plasmid treated samples (at seven days post electroporation) were quantified relative to *smat1* as described previously [64]. The Geometric mean method was used to quantify *smp_089290* transcript levels in freshly perfused male and female schistosomes as described previously [65,66]. Melting curves (dissociation curves) were generated for each qRT-PCR analysis to verify the amplification of one product only. A two-tailed student's *t*-test was used to test for significance between siRNA treatments. A Kruskal-Wallis ANOVA with Dunn post-hoc comparisons was used to test for significance between CRISPR/Cas9 plasmid treatments. A two-tailed student's *t*-test was used to test for significance between freshly perfused adult male and female worms.

## Whole mount *in situ* hybridisation (WISH)

The full coding sequence of Smp_089290 was amplified using forward (5′–ATG GCT ACC GTA CCA CCG– 3′) and reverse (5′–CTA TAA CAA TGT CTG AAA CAG TCC ATC– 3′) primer pairs. The PCR products were subsequently cloned into the pJC53.2 vector [67] using standard cloning methods as mentioned previously. These recombinant plasmids were subsequently used to generate digoxigenin-labelled riboprobes using the Riboprobe System (Promega) with SP6 or T3 RNA polymerases and digoxigenin-labelled Uracil triphosphate (Roche) [68]. Antisense treatment riboprobes (generated by SP6 polymerase) and sense control riboprobes (generated by T3 polymerase) were processed as described [68] and stored at -20°C until

needed for the WISH staining protocol. Riboprobes were used within 2 wk of their initial storage at -20˚C.

Adult male and female worms were relaxed and separated by incubation (15 min) in a 0.25% solution of the anaesthetic ethyl 3-aminobenzoate methanesulphonate (Sigma-Aldrich) dissolved in DMEM [69]. Relaxed parasites were subsequently killed in a 0.6 M solution of MgCl$_2$ (1 min) and fixed for 4 hr in 4% formaldehyde in PBSTx (PBS + 0.3% Triton X-100). Fixed parasites were dehydrated in MeOH and stored at -20˚C until needed. Parasite samples were rehydrated in 1:1 MeOH:PBSTx (5–10 min) followed by incubation in PBSTx (5–10 min). Rehydrated parasite samples were bleached in formamide bleaching solution (0.5% v/v de-ionised formamide, 0.5% v/v SSC and 1.2% w/w H$_2$O$_2$, brought to a final volume of 10 ml with diethyl pyrocarbonate H$_2$O) for 90 min, rinsed twice in PBSTx, treated with proteinase K (10 μg/ml, Invitrogen) for 45 min at room temperature and post-fixed for 10 min in 4% formaldehyde in PBSTx. Parasite samples were processed as previously described [67,68] with several modifications. Antisense treatment riboprobes and sense control riboprobes were mixed with hybridisation solution (50% v/v de-ionised formamide, 10% w/v dextran sulphate, 1.25% v/v SSC, 1 mg/ml yeast RNA, 1% v/v Tween-20, brought to a final volume of 40 ml with diethyl pyrocarbonate H$_2$O) and hybridised overnight at 52˚C. Parasite samples were transferred to fresh colorimetric developing solution consisting of alkaline phosphatase buffer (100 mM Tris pH 9.5, 100 mM NaCl, 50 mM MgCl$_2$, 0.1% v/v Tween-20, brought to a final volume of 10 ml with 10% polyvinylalcohol solution) supplemented with 4.5 μl/ml NBT (Roche) and 3.5 μl/ml BCIP (Roche). All parasite samples treated with the antisense treatment riboprobe were developed at room temperature in the dark until the desired level of purple signal was reached (male samples = 2 hr, female samples = 45 min). In parallel, all parasite samples treated with the sense control riboprobe were developed for the same length of time. Once the desired level of signal was reached, the colorimetric developing solution was removed and worms were rinsed twice in PBSTx to stop any further development from occurring. Worms were dehydrated in 100% ethanol for 5 min and subsequently submerged in 80% glycerol in 1x PBS and incubated overnight at 4˚C. Thereafter, stained worms were mounted onto microscope slides and examined with a light microscope (Leica LMD6000 Laser Microdissection Microscope).

## Single-cell RNA-Seq (scRNA-Seq) analysis

Localisation of *smp_089290* (*smnagal*) found within the 68 adult worm clusters generated from available scRNA-Seq data [70] was depicted as uniform manifold approximation and projection (UMAP) plots using Seurat V3 [71].

## α-GAL/α-NAGAL enzymatic activity measurements

Soluble worm antigen preparation (SWAP) was derived from worms (freshly perfused or electroporated with siRNAs) removed from culture after 1 wk. Worms were homogenised with a 5 mm diameter stainless steel bead (Qiagen) for 4 min at 50 Hz in 100 μl of 0.15 M McIlvaine buffer pH 4.6 [72] with EDTA-free protease inhibitors (Sigma-Aldrich) using a TissueLyser LT (Qiagen). After homogenisation, tubes were centrifuged at 21,100 x *g* for 30 min at 4˚C and SWAP was collected and quantified by the Bradford method (Sigma-Aldrich).

The enzymatic activity of SWAP derived from freshly perfused adult male and female worms, si*Luc* treated and si*Smp_089290* treated adult male and female worms was measured using 4-Nitrophenyl α-D-galactopyranoside (α-GAL colorimetric substrate) and 4-Nitrophenyl N-acetyl-α-D-galactosaminide (α-NAGAL colorimetric substrate) in separate reactions. Differing concentrations of human α-GAL (Fabrazyme, kindly provided by the Leiden University Medical Centre, LUMC) and commercially sourced α-NAGAL cloned from

*Chryseobacterium meningosepticum* and expressed in *E. coli* (New England Biolabs, Ipswich, MA) were measured with corresponding substrates.

Enzyme assays were performed in standard flat-bottomed 96 well plates (STARLAB). α-GAL/α-NAGAL assays consisted of 100 μl of α-GAL or α-NAGAL substrate dissolved in 0.15 M McIlvaine buffer (pH 4.6) with the addition of differing concentrations of α-GAL or α-NAGAL and 0.15 M McIlvaine buffer pH 4.6 to give a final volume of 125 μl per well. Concentrations of α-GAL used to generate a standard curve of activity were as follows: 0.95, 0.475, 0.19, 0.0475, 0.0095, 0.00475 and 0.0019 μg/ml. Concentrations of α-NAGAL used to generate a standard curve of activity were as follows: 20, 15, 10, 5, 2 and 1 μg/ml. SWAP reactions were set-up in the same way but α-GAL/α-NAGAL enzymes were replaced with different quantities of sample specific SWAP and incubated with both α-GAL and α-NAGAL substrates. For untreated adult male and female worms, 5 μg of SWAP was used per well to enable comparison between gender. For si*Luc* treated and si*Smp_089290* treated adult male and female worms, 6.45 μg and 1.08 to 2.44 μg of SWAP was used per well to enable comparison between siRNA treatments, respectively. Reaction proceeded for 60 min at 37˚C and terminated by the addition of 70 μl of 0.4 M glycine (pH 10.4) [72]. Final absorbances were quantified at 410 nm using a POLARstar Omega microplate reader (BMG Labtech). Absorbance values produced from different SWAP treatments were compared to α-GAL/α-NAGAL standard curves to calculate α-GAL and α-NAGAL activities (μg/ml). A two-tailed student's *t*-test was used to test for significance between siRNA treated samples or between genders.

## Motility analysis quantified by WormAssayGP2 and adult worm scoring matrix

The digital image processing-based system known as WormAssayGP2 was derived from Marcellino *et al.* [73] and implemented by us as previously described [74]. Individual wells containing up to 10 adult worms were recorded for 60 sec each day and analysed by the Lucas-Kanade algorithm. Once recording was completed, the data were quantified and stored as a *.csv* file which was further processed to calculate the mean motility for the control and treatment groups. In parallel, adult worm motility was also scored using the WHO-TDR scoring matrix [75]. All worms were scored daily from two days after electroporation until seven days after electroporation. Worms were ranked between 4–0 based on their motility; 4 = normal active/paired up, 3 = full body movement but slowed activity, 2 = minimal activity, occasional movement of head and tail only, 1 = movement in the suckers only or slight contraction of the body and 0 = total absence of motility. For both motility analyses, a General Linear Mixed-Effects Model was fitted to each dataset ('NLME' package) and statistical differences were determined by performing pairwise comparisons of the estimated marginal means of each group per time point ('EMMEANS' package) in R. Video footage of adult worms was captured using a Nexius-Zoom stereo microscope (Euromex) and edited with ImageFocus 4 software (Euromex).

## Enumeration of vitellocytes and egg volume measurements

The eggs released by adult female worms from RNAi and CRISPR/Cas9 genome editing experiments were collected and fixed in 1 ml 10% neutral buffered formalin solution (Sigma-Aldrich) for 24 hr at 4˚C. Thereafter, eggs were enumerated using a Sedgewick Rafter Counting Chamber [76]. Prior to visualisation by laser scanning confocal microscopy (LSCM), stored eggs from RNAi experiments only were immersed in PBS supplemented with DAPI (4',6-diamidino-2-phenylindole, 2 μg/ml). Fluorescence images (10 eggs per treatment) were captured on a Leica TCS SP8 super resolution laser confocal microscope fitted with a 63x (water immersion) objective using the Leica Application Suite X (LAS X). Green (egg

autofluorescence) fluorescence was visualised with an argon or diode-pumped, solid state (DPSS) laser at 488 nm. DAPI was visualised using a 405 nm blue diode laser. The number of vitellocytes (DAPI$^+$ cells) and overall volume (mapped by the green autofluorescence) for individual eggs were calculated using IMARIS 7.3 software (Bitplane). A two-tailed student's *t*-test was used to test for significance between siRNA treated samples with regard to the total number of eggs produced and individual egg volume. A Kruskal-Wallis ANOVA with Dunn post-hoc comparisons was used to test for significance between CRISPR/Cas9 plasmid treated samples with regard to the total number of eggs produced. IMARIS 7.3 software (Bitplane) was also used to create a video showing the 360˚ horizontal rotation of a representative egg from each siRNA treatment.

### Vitelline droplet staining and diphenol oxidase localisation

To visualise vitelline droplets, whole siRNA treated-adult worms were stained with 1% Fast Blue BB as described previously [77]. Diphenol oxidase activity was localised in whole siRNA treated-adult worms by incubating intact, previously frozen parasites (-80˚C) in a 0.2 M KH$_2$PO$_4$ (pH 6.7) buffer containing 4 mM 3-methyl-2-benzothiazolinone hydrazone hydrochloride hydrate (MBTH, Sigma) and 1 mM l-3,4-dihydroxyphenylalanine (L-DOPA, Sigma) as described previously [78]. After incubation for 30 min at 30˚C, the reaction was stopped with 1% acetic acid. Thereafter, all worms were mounted onto microscope slides and examined with a light microscope (Zeiss Axio Imager 2 Microscope).

## Results

### *Schistosoma mansoni* contains a single α-N-acetylgalactosaminidase (SmNAGAL)

α-N-acetylgalactosaminidase (α-NAGAL) is a member of the glycosyl hydrolase (GH) 27 family and contains both a melibiase-2 (PF16499) and a melibiase-2 C-terminal (PF17450) domain. As all enzymatically important (ligand binding and catalytic) amino acids are conserved within the melibiase-2 domain [33], we focused our interrogation of the *S. mansoni* genome (v7.0) for the presence of putative homologs that contained >70% of the residues within this domain. Our analysis revealed the presence of five schistosome members that met this criterion: Smp_170840, Smp_179250, Smp_247760, Smp_247750 and Smp_089290 (**Fig 1**).

Among these five *S. mansoni* gene products, only one (Smp_089290, highlighted yellow, **Fig 1**) contained all 13 ligand binding residues (green shaded amino acids, **Fig 1**) as well as both catalytic aspartic acid residues (light blue Ds, **Fig 1**) critical for α-NAGAL activity [32]. MS3_11280 (*S. haematobium* homolog) and EWB00_005284 (*S. japonicum* homolog) also shared these diagnostic characteristics. Furthermore, while two additional *S. japonicum* homologs (EWB00_005283 and EWB00_005285) contained both catalytic aspartic acid residues, they did not possess all 13 ligand binding residues. These five schistosome proteins, containing the essential catalytic aspartic acid residues, clustered into a separate clade discrete from other putative schistosome GH27 family members (**Fig 2**).

The predicted three-dimensional structure of Smp_089290 was derived by homology modelling and passed all stereochemical quality assessments (**S1 Fig**). Homology modelling of Smp_089290 as a monomer (HsNAGAL functions as a homodimer [32]) revealed the positioning of these 13 ligand binding residues and two catalytic residues around the putative active site (purple, **Fig 3**). Despite the melibiase-2 C-terminal domain (red, **Fig 3**) not possessing any residues involved in ligand binding and substrate cleavage mechanisms, the predicted Smp_089290 model and *H. sapiens* α-NAGAL structure are consistent across this region. Both

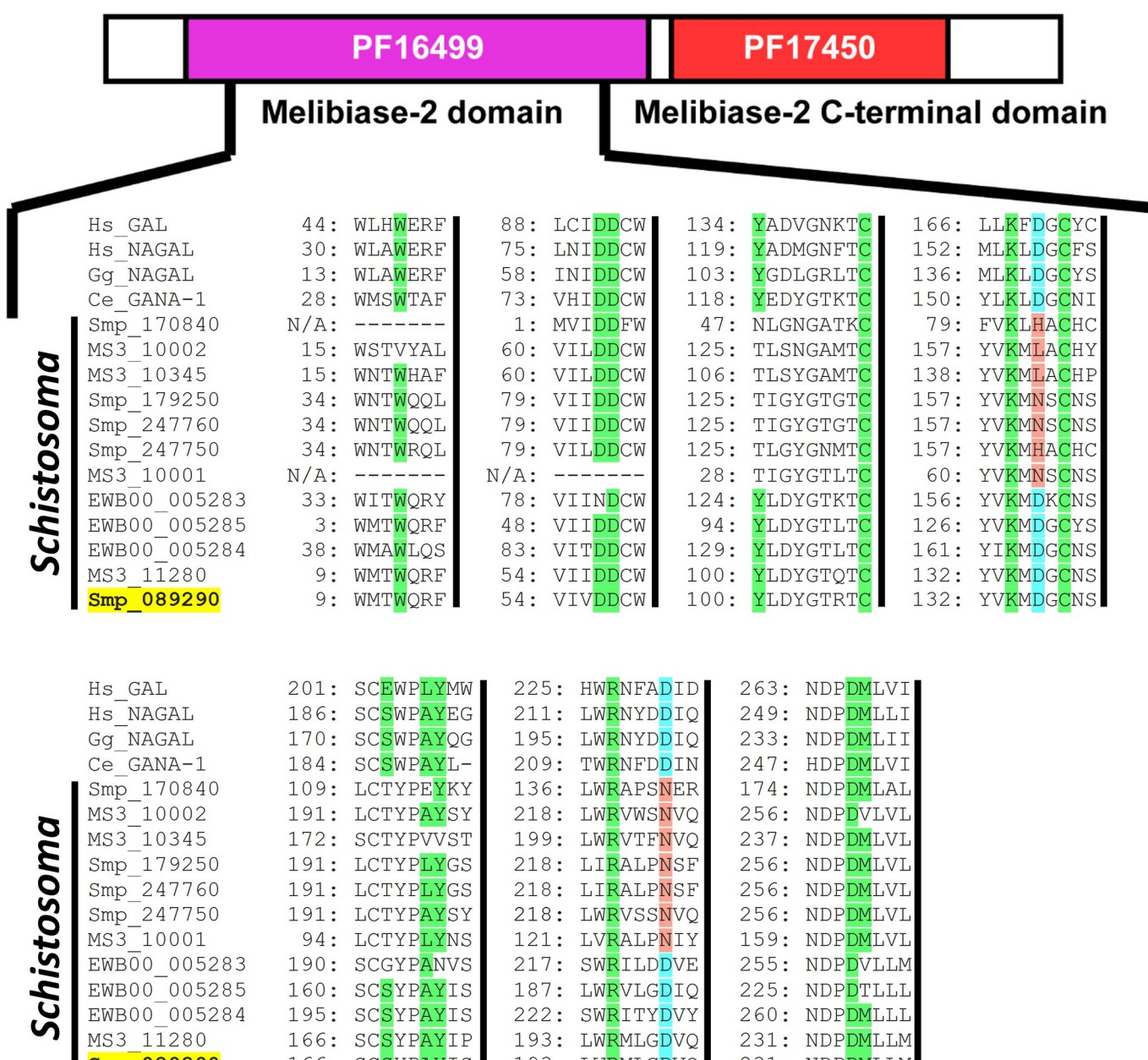

**Fig 1. *Schistosoma mansoni* contains five GH27 family members, but only Smp_089290 contains all residues necessary for α-NAGAL substrate binding and cleavage.** A concatenated multiple sequence alignment of α-GAL and α-NAGAL proteins from *H. sapiens* (Hs), *G. gallus* (Gg), *C. elegans* (Ce), *S. japonicum* (EWB), *S. mansoni* (Smp), and *S. haematobium* (MS3) throughout the melibiase-2 domain (PF16499). Numbers located at the beginning of each sequence represent the amino acid position in the protein sequence. Ligand binding residues are highlighted green whereas non-conserved amino acids in the same position in other sequences are white. Catalytic aspartic acid (D) residues are highlighted light blue whilst non-conserved amino acids in the same position in other sequences are highlighted red. Amino acid residues which are missing from 'Ce_GANA-1', 'Smp_170840' and 'MS3_10001' are indicated with — whereas N/A is used for the amino acid position.

the predicted Smp_089290 model and *H. sapiens* α-NAGAL structure possess eight anti-parallel β-strands (**Fig 3**) [33]. Overall, the homology modelling analysis predicted that Smp_089290 likely possesses folding topology and spatial arrangements that closely resemble typical α-GAL and α-NAGAL proteins.

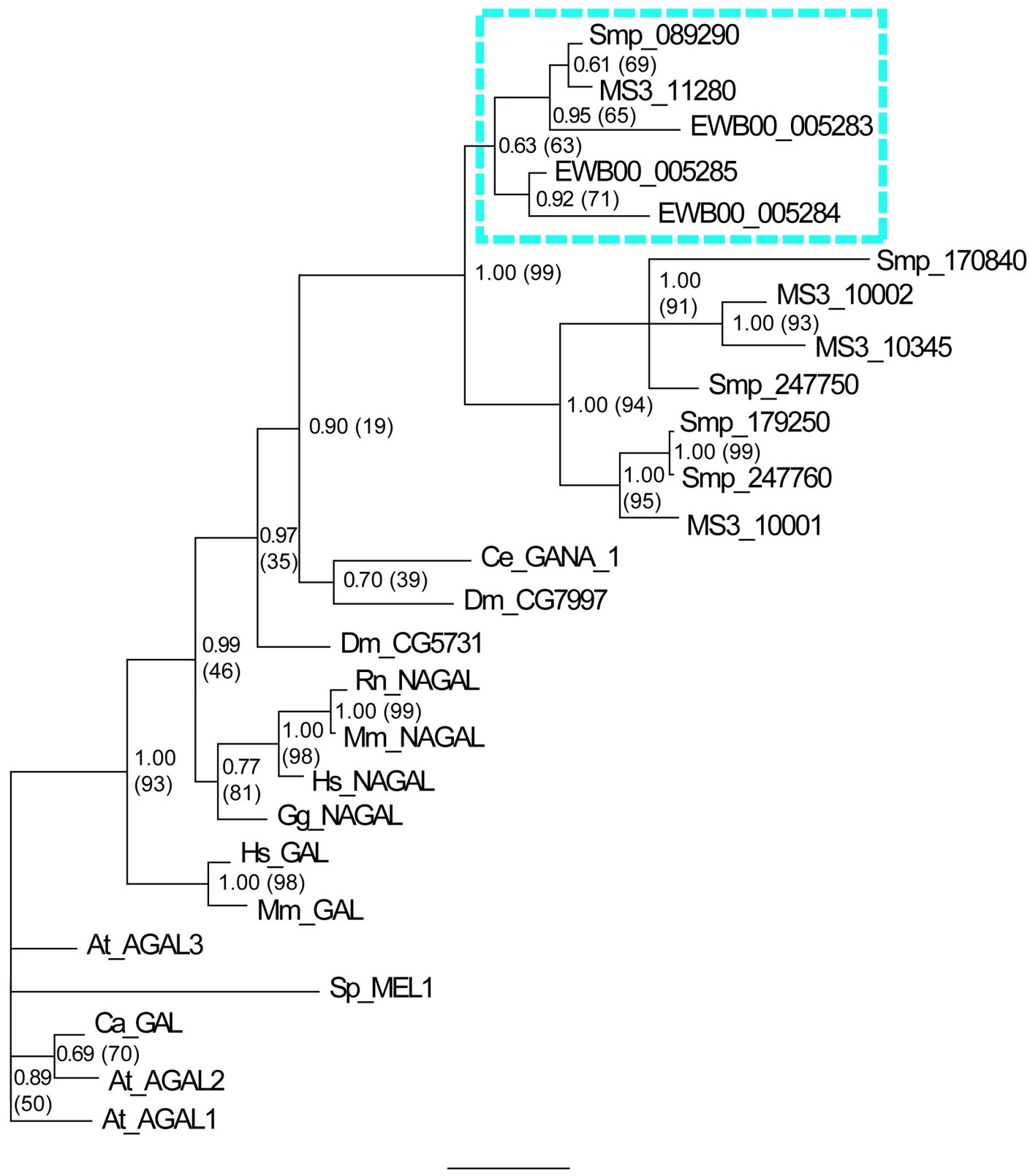

**Fig 2. *Schistosoma* melibiase-2 domain containing proteins with conserved catalytic aspartic acid residues cluster in a distinct clade of GH27 family members.** Phylogenetic analyses were conducted using a concatenated multiple sequence alignment of *Schistosoma* proteins that contained >70% of the residues within the

melibiase-2 domain and α-GAL/α-NAGAL protein types across phyla. Proteins were analysed using both Maximum Likelihood and Bayesian inference approaches. Branch lengths (indicated by scale bar) represent distance among different taxa as predicted by the Bayesian inference approach. Node labels outside parentheses represent Bayesian posterior probability support values whilst those within parentheses represent percentage bootstrap support values from Maximum Likelihood analysis. *Schistosoma* proteins conserving catalytic aspartic acid residues are highlighted in the blue dashed box. The phylogram includes protein sequences from *S. mansoni* (Smp), *S. haematobium* (MS3), *S. japonicum* (EWB), *A. thaliana* (At), *C. arabica* (Ca), *C. elegans* (Ce), *D. melanogaster* (Dm), *G. gallus* (Gg), *H. sapiens* (Hs), *R. norvegicus* (Rn), *M. musculus* (Mm) and *S. pombe* (Sp).

Together, these data suggest that the *S. mansoni* genome encodes a single gene product (Smp_089290) containing all essential catalytic amino acid residues for hydrolysis and release of α-N-acetylgalactosamine from glycosylated substrates. Therefore, this putative schistosome α-N-acetylgalactosaminidase (SmNAGAL, Smp_089290) was taken forward for further transcriptional, enzymatic, and functional genomics studies.

## *Smnagal* is developmentally regulated, female-enriched and localised to vitellaria, mature vitellocytes and parenchymal cells

To begin deciphering SmNAGAL function, both transcriptomic- and enzymatic-based approaches were initiated (**Fig 4**). Meta-analysis of historical DNA microarray data across the *S. mansoni* lifecycle [45] was facilitated by two 50-mer oligonucleotide probes that retained

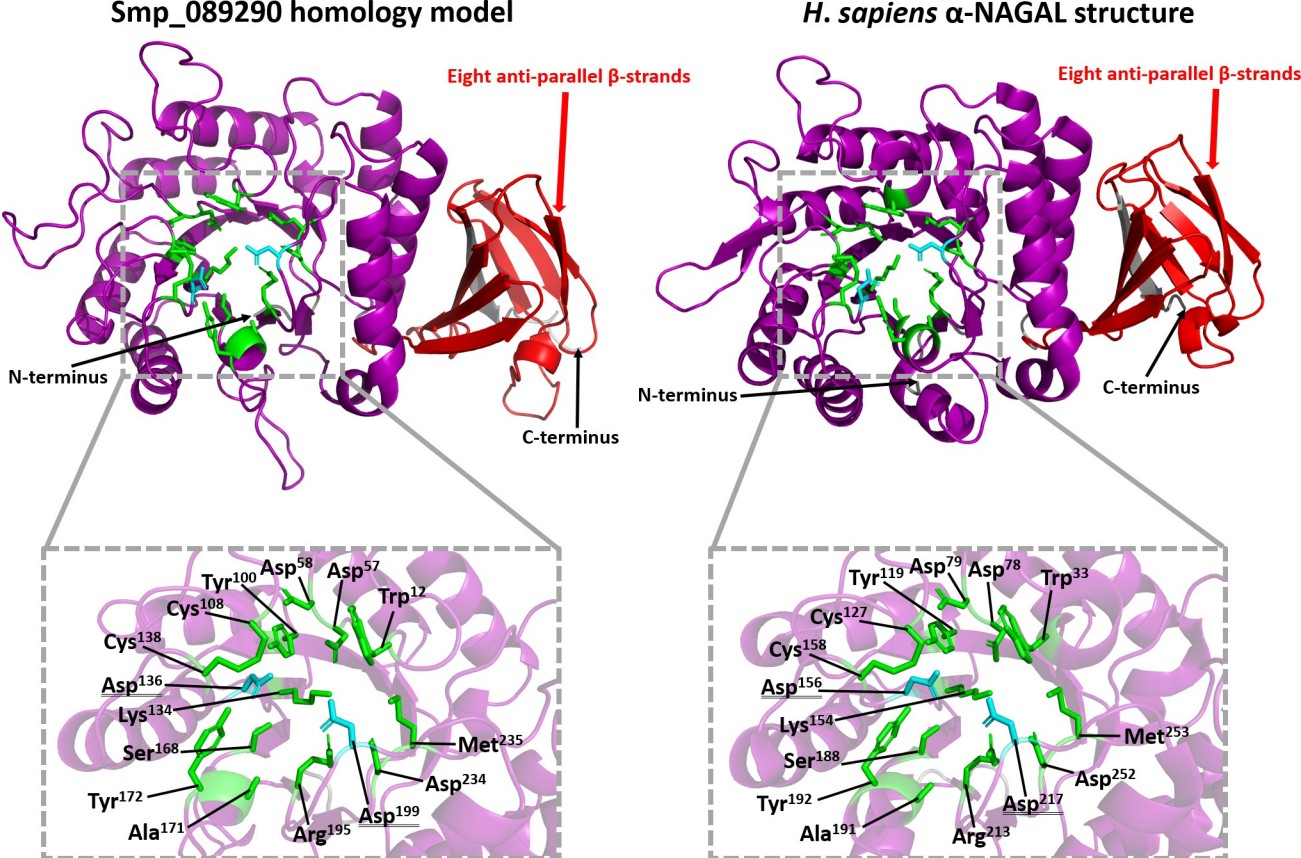

**Fig 3. Comparisons of the catalytic active site pockets found in the Smp_089290 homology model and the crystal structure of human α-NAGAL.** The grey dashed boxes depict a close-up view of the catalytic active site pocket within the Smp_089290 homology model's and *H. sapiens* α-NAGAL crystal structure's melibiase-2 domain (purple). The close-up views label all 13 ligand binding residues (green) and two catalytic Asp residues (light blue, underlined with double line) with their corresponding amino acid positions in each of the protein sequences shown. The red arrows show the location of the eight anti-parallel β-strands found near the C-terminus of the Smp_089290 homology model's and the *H. sapiens* α-NAGAL crystal structure's melibiase-2 C-terminal domain (red). Black arrows show the locations of the N and C-termini of the model/structure.

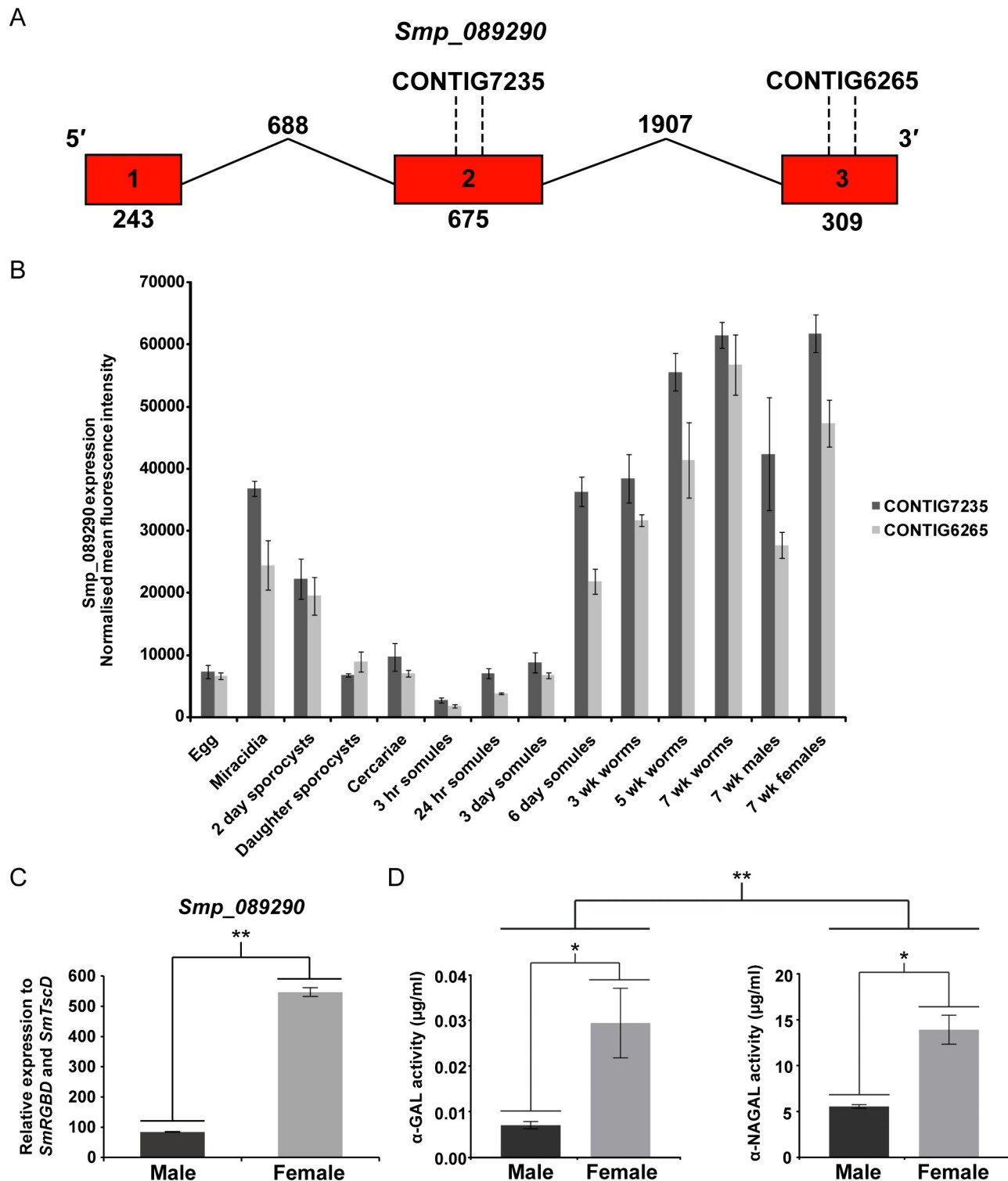

**Fig 4. smnagal (smp_089290) expression and α-NAGAL/α-GAL activities are female-enriched.** (A) Diagrammatic representation of *smnagal/smp_089290* gene structure with 50-mer oligonucleotides mapped. Exons are depicted as red boxes, which are linked by lines representing introns. Numbers written inside each exon represent their position in the gene sequence. Numbers written below exons and above introns represent their length in base pairs. The 5′ and 3′ ends are shown above exon 1 and 3, respectively. The positions of oligonucleotide 50-mers corresponding to CONTIG7235 and CONTIG6265 are shown above exon 2 and 3 respectively. (B) DNA microarray analysis of *smp_089290* expression across 15 lifecycle stages. DNA microarray gene expression profile consisted of normalised mean fluorescence intensities of *smnagal/smp_089290* transcript abundance derived from oligonucleotides CONTIG7235 and CONTIG6265 as described previously [45]. (C) *smnagal/smp_089290* transcript levels in untreated adult male and female were quantified relative to *smrgbd* and *smtscd* to validate normalised mean fluorescence intensities produced in 7 wk male and 7 wk female schistosomes. Statistical significance is indicated (Student's *t*-test, two tailed, unequal variance, ** = $p < 0.01$). (D) Equal quantities of SWAP (5 μg per

well) were used for both sexes and measured for α-NAGAL and α-GAL activity on diagnostic α-NAGAL and α-GAL substrates. Statistical significance is indicated (Student's *t*-test, two tailed, unequal variance, * = $p < 0.05$ and ** = $p < 0.01$).

100% base-pair complementarity to exon 2 (CONTIG7235) and exon 3 (CONTIG6265), respectively, of *smnagal* (**Fig 4A**). For each of these two oligonucleotides, similar patterns of *smnagal* abundance were deduced. While *smnagal* expression was low in eggs, it increased in miracidia only to wane as schistosome development (sporocysts—cercariae) continued in the molluscan host (**Fig 4B**). Upon early intra-mammalian schistosome maturation (3 hr—3 day schistosomula), *smnagal* transcription remained invariably low, until day six post schistosomula transformation. At this point and extending into more developmentally mature lifecycle forms (3 wk–7 wk schistosomes), *smnagal* expression increased, reaching peak abundance in 7 wk old schistosomes. Here, female-enriched *smnagal* expression was clearly observed in adult schistosomes, which was independently confirmed by qRT-PCR (**Fig 4C**). Additionally, female-biased expression of *smnagal* was observed when interrogating *S. mansoni* RNA-Seq meta data (**S2 Fig**) [47]. Further support for female biased expression of *smnagal* was obtained from the analysis of α-N-acetylgalactosaminidase activity in adult worm protein extracts (**Fig 4D**). In a direct comparison, female protein extracts contained significantly higher levels of α-NAGAL activity when compared to males. This female biased trend was also observed with α-GAL activity measured in the same extracts (**Fig 4D**).

Expanding these temporal investigations of *smnagal* abundance and enzymatic activities to spatial localisation, by WISH and scRNA-Seq, in adult worms revealed additional gender-specific traits. An antisense RNA probe spanning the full length coding sequence of *smp_089290* was used for each WISH localisation experiment (**Fig 5**) [68]. A negative control was prepared using a sense *smp_089290* probe (**S3 Fig**); no specific staining was observed. While the ovary (containing mature and immature oocytes) was not a rich source of *smnagal* expression, the vitellarium (and mature vitellocytes passing through the vitello-oviduct) was highly enriched for this putative α-NAGAL gene product (**Fig 5A**). These observations were supported by scRNA-Seq approaches, which showed prominent *smnagal* abundance within the vitellaria and mature vitellocytes and no expression within female gametes (**S4 Fig**). In addition, low levels of *smnagal* expression were found within the parenchyma, which was more clearly observed in the female scRNA-Seq plots (**S4 Fig**). Moderate *smnagal* expression was also observed in some clusters of neuronal cells, tegument lineage cells and muscle cells when inspecting the female scRNA-Seq plots. No appreciable expression was found in any other female tissue examined. The WISH analysis of males revealed that *smnagal* expression was predominantly localised to parenchymal cells widely distributed throughout the body (**Fig 5B**), which was confirmed by the male scRNA-Seq expression profile plots (**S5 Fig**). A lack of staining was additionally observed throughout the primary reproductive organs (testes), which was supported by scRNA-Seq profiles. WISH analyses also showed that cells lining the tegument and intestine lacked intense *smnagal* expression in both genders, although this was easier to deduce in males. Similar to females, small yet noticeable levels of *smnagal* expression were evident in some neuronal cell clusters when inspecting male scRNA-Seq plots (**S5 Fig**). Appreciable expression was not observed elsewhere in any other male tissue examined.

## α-N-acetylgalactosaminidase activity is required for worm motility, egg production and development

The localisation of *smnagal* to adult parenchymal cells, neuronal clusters and mature vitellocytes as well as the well-documented neuromuscular defects characteristic of Schindler/

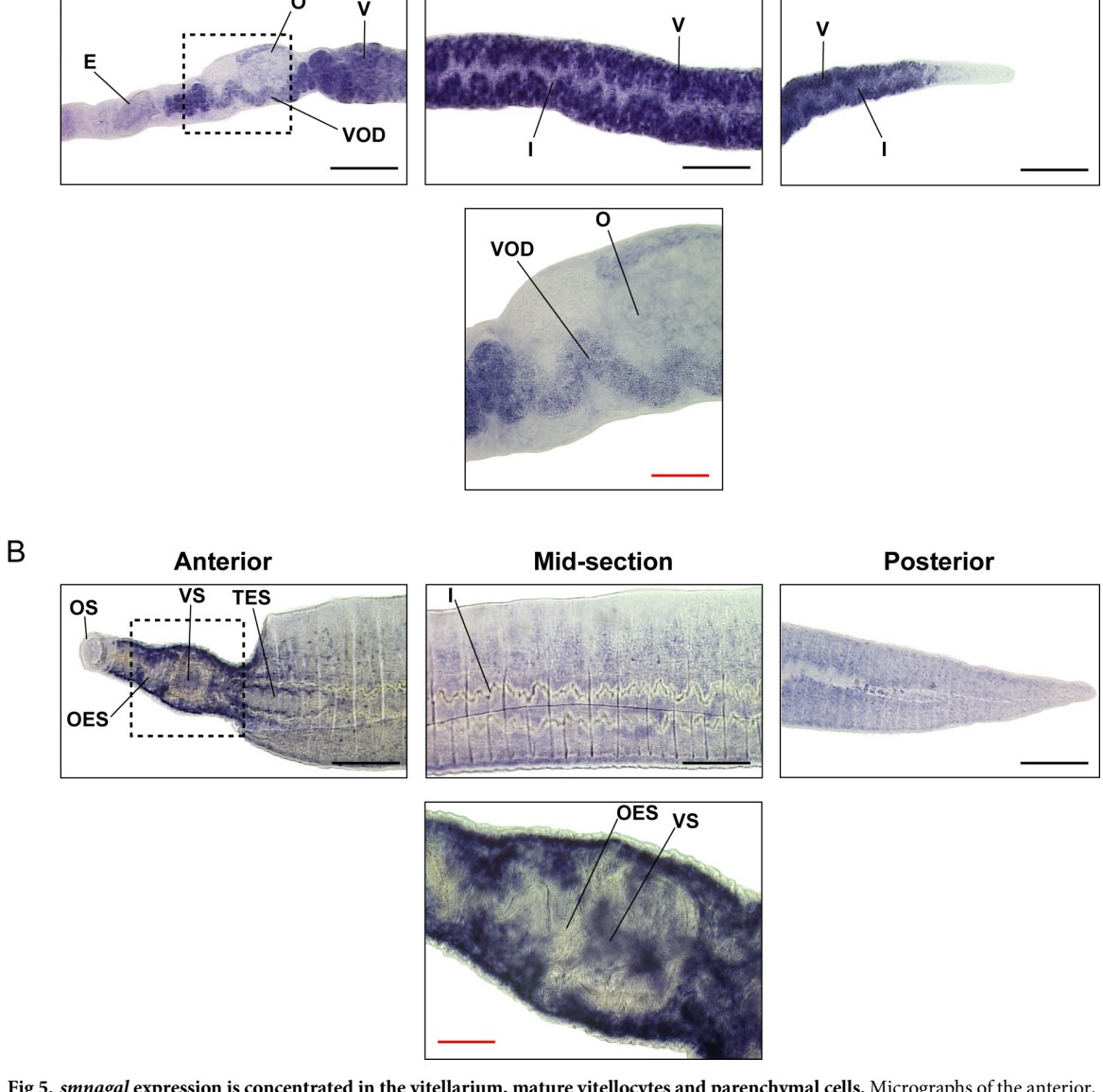

**Fig 5. *smnagal* expression is concentrated in the vitellarium, mature vitellocytes and parenchymal cells.** Micrographs of the anterior, mid-section and posterior (10x magnification) of (A) female and (B) male schistosomes as well as anterior images with a higher magnification (40x magnification, area depicted by black dashed box). Structures labelled include egg (E), ovary (O), vitellarium (V), vitello-oviduct (VOD), intestine (I), oral sucker (OS), oesophagus (OES), ventral sucker (VS) and testes (TES). Black scale bars = 200 μm and red scale bars = 50 μm.

Kanzaki disease (due to *α-nagal* deficiencies, [23,24]) implicated key roles for this gene product in schistosome motility, oviposition and development. Therefore, to assess whether these processes were dependent upon α-NAGAL activity, functional genomics investigations of *smnagal/smp_089290* were conducted in adult schistosomes (**Fig 6**). RNA interference (RNAi) of *smnagal/smp_089290*, using small interfering RNAs (siRNAs), led to a highly significant knockdown (92%) of *smnagal* in adult male worms when compared to controls (**Fig 6A**). In parallel, genome editing approaches were implemented with CRISPR/Cas9 plasmid constructs and used to complement RNAi experiments. The CRISPResso2 pipeline was utilised to

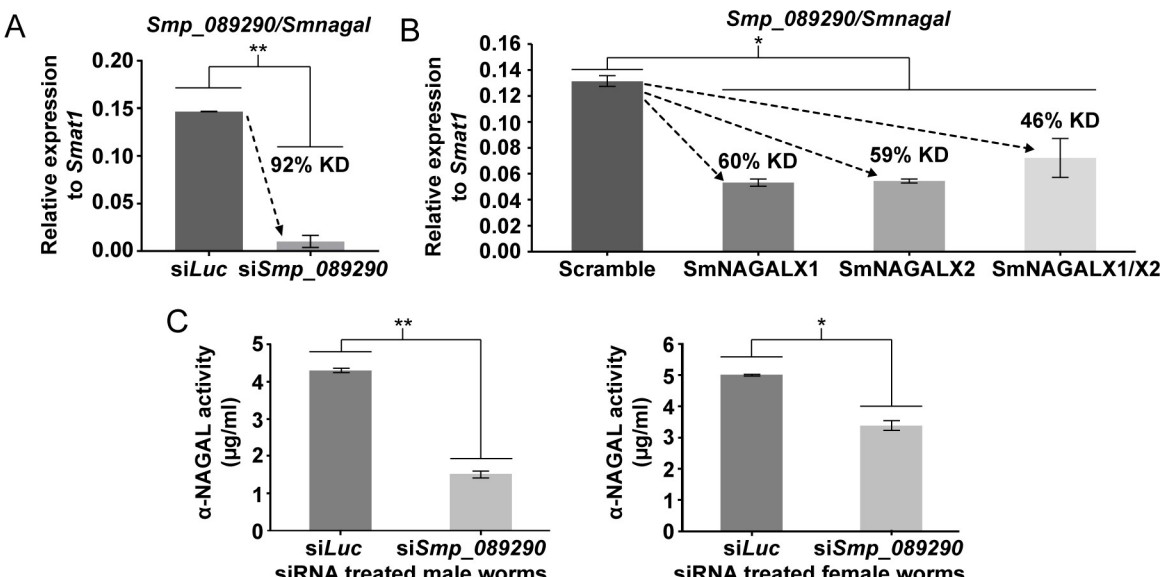

**Fig 6. RNAi and CRISPR/Cas9 approaches lead to *smnagal* knockdown and siRNA-treated adult worms contain reduced α-NAGAL activity.** (A) 7 wk old adult male schistosomes were electroporated with 5 μg siRNA duplexes targeting luciferase (si*Luc*) and *smp_089290* (si*Smp_089290*). After 48 hr, total RNA was isolated and used to generate cDNA, which was subjected to qRT-PCR. Percent knockdown (KD) is indicated. Statistical significance is indicated (Student's *t*-test, two tailed, unequal variance, ** = *p*<0.01). (B) 7 wk old adult male and female schistosomes were electroporated with either lentiviral CRISPR/Cas9 plasmid constructs targeting exon 1 (SmNAGALX1) or exon 2 (SmNAGALX2) of *smnagal*. Additionally, both exons were targeted by electroporating a mixture of both plasmid constructs (SmNAGALX1/X2). Electroporations with CRISPR/Cas9 plasmid DNA containing a Scramble sgRNA were used as a control. After seven days, total RNA was isolated and used to synthesise cDNA for qRT-PCR analysis. Percent knockdown (KD) is indicated. Statistical significance is indicated (Kruskal-Wallis ANOVA with Dunn post-hoc comparisons, * = *p*<0.05). (C) α-NAGAL activity was measured in SWAP derived from si*Luc* treated and si*Smp_089290* treated adult male and female worms (6.45 μg for males and 2.44 μg for females) as described above. Statistical significance is indicated (Student's *t*-test, two tailed, unequal variance, * = *p*<0.05 and ** = *p*<0.01).

quantify NHEJ-associated mutations within aligned MiSEQ library sequencing reads from genome-edited worms (**S1 Table**). Oligonucleotide primers designed for MiSEQ library generation were designed to encompass the programmed DSB sites for each sgRNA used (**S6 Fig**). All genome edited samples showed detectable levels of genome editing (0.25–0.31%) with substitutions being the most frequently observed mutation (**S2 Table**). Further analyses of modified sequence reads revealed all CRISPR/Cas9 plasmid treatment samples displayed NHEJ-associated mutations predicted to introduce frameshifts at the *smnagal* locus (likely resulting in the translation of substantially truncated proteins) or to ablate *smnagal* transcription (**S7 Fig**). Similar to RNAi, CRISPR/Cas9 genome editing led to significant knockdowns (46–60%) of *smnagal* in mixed-sex adult worms when compared to controls (**Fig 6B**). Reassuringly, *smnagal* depletion in siRNA-treated worms significantly reduced α-NAGAL activity in SWAP derived from both males and females when compared to si*Luc* controls (**Fig 6C**). However, RNAi-mediated *smnagal* knockdown did not result in a significant reduction in SWAP-derived α-GAL activity when compared to si*Luc* treated schistosome samples (**S8 Fig**).

Having established that *smnagal* encodes a functional α-NAGAL and given that RNAi and CRISPR/Cas9 depleted this transcript from intracellular RNA pools, motility and egg-laying phenotypes of si*Smnagal* treated schistosome pairs were subsequently examined and quantified. Regardless of the quantification metric used (WormAssayGP2 [73,74] or WHO-TDR standards [75]), a clear motility defect was observed in both male and female schistosomes when *smnagal* was depleted by RNAi (**Figs 7A** and **S9** and **S1 Video**). This motility defect was apparent by day two post RNAi and maintained until day 7, when the assay was terminated.

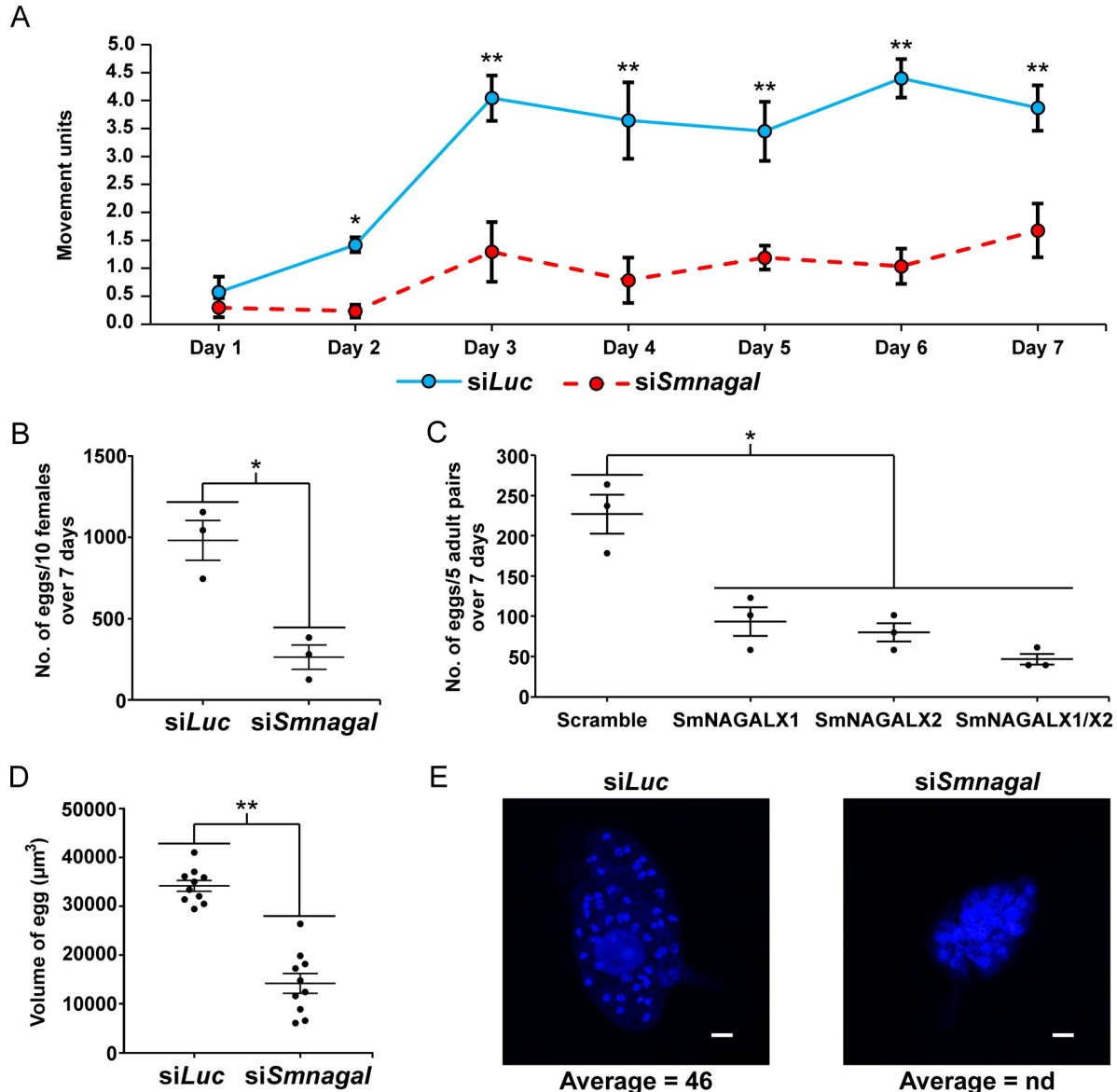

**Fig 7. Reductions in *smnagal/smp_089290* affect adult worm motility and egg production processes.** (A) Motility of si*Luc* treated and si*Smnagal* treated adult male and female worms was analysed daily for up to seven days after electroporation using WormAssayGP2 as described in the Materials and methods. Statistical significance is indicated (General Linear Mixed-Effects Model, NLME and EMMEANS R packages, * = $p < 0.05$ and ** = $p < 0.01$). (B) The total number of eggs produced by si*Luc* treated and si*Smnagal* treated adult female worms were collected seven days after electroporation and enumerated. Statistical significance is indicated (Student's *t*-test, two tailed, unequal variance, * = $p < 0.05$). (C) The total number of eggs produced by adult worm pairs electroporated with SmNAGALX1 CRISPR/Cas9 plasmid, SmNAGALX2 CRISPR/Cas9 plasmid and SmNAGALX1/X2 CRISPR/Cas9 plasmid were counted seven days after electroporation. Electroporations with lentiviral CRISPR/Cas9 plasmid constructs containing a Scramble sgRNA served as a control. Statistical significance is indicated (Kruskal-Wallis ANOVA with Dunn post-hoc comparisons, * = $p < 0.05$). (D) Volumes of eggs produced by si*Luc* treated and si*Smnagal* treated adult female worms were calculated as described in the Materials and methods. Statistical significance is indicated (Student's *t*-test, two tailed, unequal variance, ** = $p < 0.01$). (E) Representative images of fluorescence in eggs collected from wells of si*Luc* and si*Smp_089290* treated worm pairs. Blue = DAPI+ cells (405 nm blue diode laser) and white scale bars = 20 µm. Average number of DAPI+ cells per egg calculated by IMARIS 7.3 software for each siRNA treatment is shown below each image. DAPI+ cells in eggs derived from wells of si*Smp_089290* treated worm pairs could not be determined and are labelled as 'nd'.

Upon completion of the RNAi assays (day seven), eggs were collected from *in vitro* cultures and quantified for number, volume and retention of mature vitellocytes. Here, a significant reduction in the quantity of deposited eggs was associated with *smnagal* deficiency (**Fig 7B**). The CRISPR/Cas9 genome editing approach further supported these observations with all *smnagal*-edited worms displaying a significantly reduced number of deposited eggs when compared to controls (**Fig 7C**). Notably, eggs derived from siRNA-treated female worms were also significantly smaller than those collected from wells of si*Luc* treated worms (**Fig 7D**) and contained abnormally spaced vitellocytes (**Figs 7E** and **S10** and **S2 Video**).

Mature vitellocytes contain two types of large cytoplasmic inclusions known as vitelline droplets and lipid droplets, which both coalesce to assist in eggshell formation [79,80]. Furthermore, eggshell formation also requires phenol oxidase-mediated protein cross-linking (quinone tanning) facilitated by tyrosinase activity originating from mature vitellocytes [77,79,81–83]. To further investigate the potential role of SmNAGAL in vitellaria biology and egg-laying, si*Luc* treated and si*Smnagal* treated adult female worms were either stained with Fast Blue BB (to visualise vitelline droplets in mature vitellocytes) or used for diphenol oxidase localisation (to assess tyrosinase activity in mature vitellocytes) assays and imaged by bright-field microscopy. In both assays, no differences were observed between siRNA treatments (**S11 Fig**).

## Discussion

The identification and inhibition of gene products responsible for essential developmental or gender-associated processes provide a pathway by which schistosome drug discovery can progress rationally. To fast track such investigations, an extensive collection of putative *S. mansoni* drug candidates is currently available within the TDR Targets database [84]. Alongside this resource, multiple reports have recently described how genome sequencing outputs can be effectively leveraged by both cheminformatics and functional genomics for characterising next-generation schistosome drug targets and chemotherapeutics [74,85–90]. Complementary evidence is provided here to support an essential role for SmNAGAL in the regulation of worm movement and reproductive processes.

SmNAGAL (Smp_089290) encoded by *S. mansoni*, MS3_11280 encoded by *S. haematobium* and EWB00_005284 encoded by *S. japonicum* were the only melibiase-2 domain-containing *Schistosoma* proteins conserving all functionally important amino acid residues necessary for the hydrolysis of α-galactose and α-N-acetylgalactosamine residues from glycolipid and glycoproteins (**Fig 1**). Closer inspection of these residues showed these three proteins possessed the exact same ligand binding residues to *H. sapiens* α-NAGAL, *G. gallus* α-NAGAL and *C. elegans* gana-1, which are all α-NAGAL enzymes with hydrolytic activity against terminal α-N-acetylgalactosamine and α-galactose moieties [25,32,33]. Accordingly, it is likely that Smp_089290, MS3_11280 and EWB00_005284 would exhibit enzymatic activity towards both terminal α-N-acetylgalactosamine and α-galactose residues unlike α-GAL enzymes, which can only cleave terminal α-galactose residues. Studies characterising human α-GAL/α-NAGAL activity provide evidence that *H. sapiens* α-GAL cannot use α-N-acetylgalactosamine as a substrate due to steric hindrance mediated by Glu$^{203}$ and Leu$^{206}$ [31,32]. In contrast, *H. sapiens* α-NAGAL, Smp_089290, MS3_11280 and EWB00_005284 all possess Ser and Ala at homologous positions (Ser$^{188, 168, 168, 197}$ and Ala$^{191, 171, 171, 200}$), which are required/essential for using α-N-acetylgalactosamine as a substrate [32]. Additionally, EWB00_005283 and EWB00_005285 encoded by *S. japonicum* contain both catalytic aspartic acid residues (**Fig 1**), which suggests that *S. japonicum* possesses three melibiase-2 domain-containing proteins capable of enzymatic activity. However, neither of these *S. japonicum* homologs possess all 13 ligand binding residues, which may influence affinity to target substrates. Although mutagenesis studies

performed on *Pichia pastoris* α-GAL and α-NAGAL showed Trp[16] to be essential for enzymatic activity [91], little is known on the essentiality of the other ligand binding residues. Nevertheless, phylogenetic analyses reinforced that Smp_089290, MS3_11280, EWB00_005284, EWB00_005283 and EWB00_005285 are schistosome α-NAGALs as all demonstrated stronger relations to representative α-NAGAL proteins compared to representative α-GAL proteins (**Fig 2**). Whether the other GH27 family members identified here contribute to the α-GAL activities measured within adult worm extracts (**Fig 4D**) has yet to be determined.

Interrogating DNA microarray (**Fig 4B**) and RNA-Seq meta-analysis (**S2 Fig**) databases provided the first insight to *smnagal*'s temporal expression profile across the developmental stages of the schistosome. In both cases, expression of *smnagal* increased throughout intramammalian schistosome development until full adult worm maturation, suggesting a potential role for SmNAGAL in adult schistosome development within the definitive human host. Confirmation of *smnagal* expression in adult male and female worm stages by qRT-PCR analyses (**Fig 4C**) and dominant α-NAGAL activity as shown by enzymatic assays (**Fig 4D**) suggests SmNAGAL preferentially cleaves off α-N-acetylgalactosamine (and not α-galactose) residues from glycan substrates (functionally confirmed by RNAi, **Figs 6** and **S8**). Homology modelling suggests SmNAGAL possesses a pattern of α-helices and β-strands throughout its N-terminal domain (**Fig 3**) comparable to the $(\beta/\alpha)_8$ barrel structure commonly observed in GH27 family members [31–33,92–94]. Furthermore, all enzymatically important residues were shown to be arranged in an exposed catalytic active pocket, which suggests SmNAGAL utilises a double displacement mechanism for substrate binding and cleavage (**Fig 3**). This type of cleavage mechanism is commonly utilised by retaining GH enzymes (yielding a product that possesses the same anomeric configuration as the cleaved substrate) and involves two nucleophilic attacks on the 1-carbon of the substrate [33,95–98].

Amongst a variety of glycans and glycoconjugates in adult schistosomes is the O-glycopeptide Galβ1-3GalNAcα1-Ser/Thr (also known as the oncofetal Thomsen-Friedenreich antigen or TF antigen), which is an abundant α-N-acetylgalactosamine-containing structure [99]. The TF antigen is present on the surface syncytium and may be involved in protecting tegumental structures that are essential for schistosome survival within the vasculature of the human host [99–101]. Furthermore, the TF antigen has been suggested to interfere with the functions of host Kupffer cells and hepatocytes [99,102]. Therefore, SmNAGAL activity required for hydrolysis of the GalNAcα1-Ser/Thr linkage during O-glycopeptide degradation may have an impact on adult worm tegument metabolism and host interactions. Expression of *smnagal* in adult female and male tegument lineage cells identified by scRNA-Seq analyses (**S4** and **S5 Figs**) further supports a potential role for SmNAGAL in tegument metabolism and host interactions. Parenchymal expression of *smnagal* (**Fig 5**) and noticeable abundance in other cell types might be explained by the reported localisation of α-NAGAL enzymatic activity in lysosomes [103]. Lysosomes are found throughout many different *S. mansoni* tissues [104], which suggests other functional roles for SmNAGAL in addition to those identified in this study. Clearly, these observations require further exploration.

Knockdown of *smnagal* correlated with striking motility defects (**Figs 7A** and **S9**) as early as day two post siRNA treatment. Smaller reductions in motility observed in si*SmNAGAL* treated female worms may be due to the higher expression of *smnagal* in female worms compared to male worms. Higher mRNA turnover rates and transcript abundances have been shown to be a limiting factor of siRNA efficiency when targeting DGKE and ARHGAP27 kinases in HeLa and HepG2 cells [105]. Regardless of this potential discrepancy of RNAi efficiency between the sexes, the motility defect became more severe at day three (**Figs 7A** and **S9** and **S1 Video**) and was maintained for the seven day experiment. The observed abnormal motility defects were consistent with the neurological and neuromuscular impairments

associated with the human lysosomal storage disorder known as Schindler/Kanzaki disease (human α-NAGAL deficiency) [23,24,32]. Symptoms of Schindler/Kanzaki disease include a wide range of clinical neurological/neuromuscular deficits due to the accumulation of substrates possessing α-N-acetylgalactosamine residues, which are grouped into three distinct types [32]. The most severe form, type I, is characterised by stiff movements (spasticity) caused from involuntary muscle spasms, developmental retrogression, decorticate posturing, profound psychomotor retardation and muscular hypotonia, which begins in infancy [24,106–108]. Worms substantially depleted of *smnagal*/SmNAGAL activity display motility defects consistent with the spasticity associated with type I Schindler/Kanzaki disease. Due to the neurological nature of Schindler/Kanzaki disease, the onset of impaired motility phenotypes may be due to *smnagal* depletion within neuronal cell clusters in both adult female and male worms (scRNA-Seq expression profiles; **S4** and **S5 Figs**). Similarly, female muscle cells also possess moderate *smnagal* levels, which suggests the onset of RNAi-mediated motility phenotypes could be driven by depletion in muscle cell activity and, therefore, collectively characterised as a neuromuscular impairment. However, minimal *smnagal* expression observed in male muscle cells suggests the abnormal motility phenotype is exclusively associated with neurological impairments in this sex. Regardless of the differential molecular mechanisms involved, it is clear that SmNAGAL also contributes to coordinated movement in adult schistosomes.

A fundamental key difference between the male and female schistosome centres on production of eggs, which leads to the pathology and transmission of schistosomiasis [109,110]. An important organ necessary for egg production is the vitellarium, which extends throughout the majority of the female worm and is involved in the production of mature vitellocytes [80,111]. Vitellocyte maturation within the vitellarium progresses through four stages involving cell division and differentiation to ultimately produce mature vitellocytes (also referred to as stage 4 vitellocytes) [112]. Mature vitellocytes are transported through the vitello-oviduct and surround the ovum, which is initially produced in the ovary [113–115]. The rigid insoluble eggshell is subsequently synthesised by phenol oxidase-mediated protein cross-linking (quinone tanning) as a result of increased tyrosinase activity originating from late/mature vitellocytes [77,79,81–83]. Subsequently, the egg enters the uterus and is expelled through the gonopore of the female into the blood [83]. The elevated expression of *smnagal* within the vitellaria and mature vitellocytes traversing the vitello-oviduct (**Figs 5A** and **S4**) suggest SmNAGAL may be involved in aspects of egg production. The detection of *smnagal* expression within the egg (**Figs 4B** and **S2**) may also be explained by the presence of mature vitellocytes and strongly supports a role for SmNAGAL in vitellogenesis and oviposition, which was subsequently confirmed by both RNAi and programmed gene knockout (**Fig 7B** and **7C**). Here, diminished numbers of eggs produced by *smnagal* depleted adults exhibited two predominant abnormalities; lack of typical vitellocyte structuring and spacing (**Figs 7E** and **S10** and **S2 Video**) and reductions in egg volume (**Fig 7D**). *smnagal* deficiency, however, does not influence vitellocyte incorporation as DAPI⁺ cells are observed in all eggs obtained from si*SmNAGAL* treatment groups. Reductions in SmNAGAL enzymatic activity (**S11A Fig**) also does not appear to influence the presence of vitelline droplets (**S11B Fig**) or the activity of tyrosinase (**S11C Fig**) in mature vitellocytes (within the vitellaria). While both of these features (along with lipid droplets) contribute to the formation of hardened/tanned eggs [77,79–83], our data would suggest that their presence is unaffected by *smnagal* knockdown. Instead, SmNAGAL activity seems to affect an area of egg-laying biology (involving the vitellaria and vitellogenesis) that is currently unknown, but is associated with normal spacing of mature vitellocytes within the *in vitro* laid eggs (IVLEs) and total egg volume (i.e. si*SmNAGAL* treatment groups—approximately 15,000 μm³ per egg; si*Luc* treatment groups—approximately 35,000 μm³ per egg; **Fig 7D**). In some cases, *smnagal* deficiency also contributed to additional observed phenotypes such as

abnormal shaped eggs and incomplete development of the lateral spine (**S10 Fig**). Therefore, in addition to adult worm motility, SmNAGAL clearly participates in vitelline droplet/tyrosinase-independent mechanisms of oviposition. In light of this evidence, *S. japonicum* may need three enzymatically active α-NAGAL proteins (EWB00_005284, EWB00_005283 and EWB00_005285, **Fig 1**) due to markedly higher rate of oviposition (>2000 eggs per day per worm pair) compared to *S. mansoni* and *S. haematobium* (>300 and >200 eggs per day per worm pair, respectively) [116,117]. Further characterisation of these other schistosome homologs could provide further insight into the differential rate and absolute numbers of eggs produced by the three main, human-infecting schistosome species.

The use of RNAi to characterise *S. mansoni* genes of interest [74], such as *smnagal*, has been established for several years and is currently considered the functional genomics gold standard for this parasite. However, CRISPR/Cas9 genome editing approaches to characterise schistosome gene function are increasingly being explored as recently exemplified by studies of *S. mansoni omega-1* [118], *ache* [119], and *sult-or* [120]. Our use of this technology to edit *smnagal* contains some broadly-overlapping similarities to these previous studies. For example, the overall percentages of modified sequence reads observed in *smnagal*-edited worms (0.25–0.31%) were comparable to those observed in *omega-1*-edited eggs (approximately 4.5%) [118], *ache*-edited eggs (0.0295–0.12%) [119], *sult-or*-edited worms (0.3–2.0%) and *sult-or*-edited sporocysts (0.1–0.2%) [120]. Furthermore, the predominant types of mutation in *smnagal*-edited worms were substitutions (**S2 Table**), which is consistent with *omega-1*-edited eggs [118] and *ache*-edited eggs [119]. The CRISPR/Cas9 investigation targeting the liver fluke granulin (*ov-grn-1*) locus in *Opisthorchis viverrini* also showed substitutions (98.7%) to be introduced at a higher rate than insertions (0.6%) or deletions (0.7%) [121]. However, this observation was not consistent for the *sult-or* investigation, which only showed deletions attributable to genome editing [120]. Additional investigations are necessary in *S. mansoni* and other platyhelminths to confirm if predominant substitution rates resulting from programmed genome editing are a conserved feature. It is notable that worms treated with CRISPR/Cas9 plasmids targeting *smnagal* exon 1 (SmNAGALX1 and dual SmNAGALX1/X2) exhibited modified sequence reads with complex rearrangements consisting of insertions with deletions (**S2 Table**), which was not previously reported in the other *S. mansoni* genome editing studies to date. This may be due to the infrequency of this type of indel being introduced when DSBs are repaired after CRISPR/Cas9 genome editing [122]. Nonetheless, low percentages of *S. mansoni* genome editing quantified within *omega-1*-edited eggs and *ache*-edited eggs led to pronounced phenotypes [118,119]. This observation was also consistent for *smnagal*-edited worms, which showed significant reductions in IVLE production (**Fig 7C**) similar to si*SmNAGAL* treated worms (**Fig 7B**). Further comparisons can be made between *smnagal*-edited worms and *omega-1*-edited eggs in which both manipulated groups showed significant reductions in target transcript abundance. However, this is in contrast to the CRISPR/Cas9 study targeting *sult-or*, which showed no mRNA knockdown or expected phenotypes despite NHEJ-associated deletions predicted to cause frameshifts that ablate *sult-or* transcription [120]. Collectively, these results suggest that phenotypic effects/reductions in mRNA abundance associated with RNAi-treated *S. mansoni* parasites targeting a particular transcript may not always be equivalent to those found in CRISPR/Cas9-edited *S. mansoni* parasites that target the associated gene locus. Furthermore, the overall percentages of genome editing may be underestimated in *smnagal*-edited worms due to the presence of large deletions completely removing the primer regions over exon 1 and 2 and, thus, some mutations remain undetected by amplicon sequencing and bioinformatics analysis of the alleles. This situation has been reported with *S. stercoralis* [123] and *C. elegans* [124] and speculated to occur in the reports on *omega-1* [118], *sult-or* [120] and *ache* [119].

In addition to the presence of large deletions, the *omega-1* investigation suggested that several non-synonymous substitutions may have disrupted the ribonuclease catalytic site and contributed to the mutant phenotypes. Similarly, the *smnagal* genome edited-associated phenotypes observed (*smnagal* knockdown and reduced egg production) may also be due to NHEJ-associated indels/substitutions producing frameshifts (leading to the translation of substantially truncated proteins), ablating *smnagal* transcription or leading to translated proteins that lack all the functional amino acid residues (**S7 Fig**) [33,125]. The continued refinement of CRISPR/Cas9 technology in *S. mansoni* will help resolve some of these outstanding queries.

While the specific SmNAGAL targets within adult schistosomes have yet to be identified, our functional characterisation of this glycogene product suggests that glycoproteins/glycolipids containing α-N-acetylgalactosamine residues are critical for coordinated worm movement and egg production. These traits suggest that SmNAGAL is an essential schistosome gene product, representing a novel parasite vulnerability for exploiting further as a next-generation anthelminthic target for controlling schistosomiasis. As an important component of these investigations, we additionally confirm that *smnagal* is susceptible to somatic genome editing and contribute to the growing literature on utilising the CRISPR/Cas9 system in *S. mansoni* as a tool for functional genomics in parasitic platyhelminths [126]. An enhanced understanding of SmNAGAL or other *S. mansoni* glycan machinery components in lifecycle functions or host interactions will aid the search for urgently-needed, next-generation interventions.

## Supporting information

**S1 Fig. Assessing the stereochemical quality of the Smp_089290 homology model using RAMPAGE Ramachandran plot analysis, ProSA-web and Verify 3D software.** (A) The graphical representation of the RAMPAGE Ramachandran plot analysis for the Smp_089290 homology model. The plot depicts the torsional angles (phi, φ, x-axis and psi, Ψ, y-axis) of the amino acid residues in the homology model; this illustrates which combinations of angles for each atom is possible by considering their dimensions and van der Waals radii. Stable and unstable conformations of the model can, therefore, be plotted on the graph. (B) The graphical representation of the ProSA-web analysis for the Smp_089290 homology model. The z-score (-7.44, black dot) indicates the overall model quality, which is displayed in a plot that contains the z-scores of all experimentally determined protein chains in the current PDB database. The input structure is verified when it is found to be within the range of z-scores typically found for deposited proteins of similar size. (C) The graphical representation of the Verify 3D analysis for the Smp_089290 homology model. The analysis determines the compatibility of the atomic model (3D) of the input structure with its own amino acid sequence (1D) by assigning a structural class for each amino acid residue based on its location and environment (as part of an α-helix, β-strand or an interconnecting loop) and polarity. (D) A summary table of the analysis tool used (colour coded: RAMPAGE Ramachandran plot analysis = red, ProSA-web = orange and Verify 3D = yellow), the results obtained from the assessment of the Smp_089290 homology model and the expected values for verified structures.
(TIF)

**S2 Fig. Quantification of *smp_089290* abundance across the *S. mansoni* lifecycle by RNA-Seq analysis reinforces female-biased expression.** RNA-Seq meta data analysis of smp_089290 abundance for 11 lifecycle stages during mixed-sex infections. Individual expression values for male and female are only plotted for lifecycle stages where expression was assessed in sex-separated samples by Protasio et al. [51] and Lu et al. [52] (i.e. from '21 day juveniles' to '>42 day adults'). A single expression value is plotted for lifecycle stages where expression was assessed in mixed-sex samples by Anderson et al. [48], Wang et al. [49] and

Protasio et al. [50] (i.e. from 'egg' to '24 hr somules').
(TIF)

**S3 Fig. Adult female and male schistosomes showed little to no specific staining when hybridised with the sense *smp_089290* probe.** Images of the anterior, mid-section and posterior (10x magnification) of (A) female and (B) male schistosomes as well as anterior images with a higher magnification (40x magnification, area depicted by black dashed box). Structures labelled include egg (E), ovary (O), vitellarium (V), vitello-oviduct (VOD), intestine (I), oral sucker (OS), oesophagus (OES), ventral sucker (VS) and testes (TES). Black scale bars = 200 μm and red scale bars = 50 μm.
(TIF)

**S4 Fig. scRNA-Seq expression profile of *smp_089290* in adult female *S. mansoni* shows localisation to vitellocytes, parenchymal cells and neuronal cells.** Labelled UMAP projection plot of various cell clusters highlighted by black dashed regions. *smp_089290* scRNA-Seq expression values shown in this plot were generated from sexually mature adult female samples only. Expression values are normalised to a scale of 0–100 and colour coded (blue = low, red = high). Mature vitellocytes and late vitellocytes are labelled by red arrows.
(TIF)

**S5 Fig. scRNA-Seq expression profile of *smp_089290* in adult male *S. mansoni* shows enrichment in parenchymal and neuronal cells.** Labelled UMAP projection plot of various cell clusters highlighted by black dashed regions. *smp_089290* scRNA-Seq transcript expression values shown in this plot were generated from adult male samples only. Expression values are normalised to a scale of 0–100 and colour coded (blue = low, red = high).
(TIF)

**S6 Fig. Flanking oligonucleotide primers designed for MiSEQ library generation encompass DSB site of SmNAGALX1 and SmNAGALX2 sgRNAs.** Diagrammatic representation of the genomic sequence of *smnagal* from nucleotide (nt) positions 1500–4000 (represented by double lines) based on WormBase ParaSite entry [27]. Exons are depicted as red boxes with nt positions indicated above 5′ and 3′ ends. Numbers written inside each exon represent their position in the gene sequence. Positions and nt sequences of SmNAGALX1_sgRNA and SmNAGALX2_sgRNA are shown below exon 1 and 2, respectively (represented by red arrows). DSB sites for SmNAGALX1_sgRNA and SmNAGALX2_sgRNA are indicated by yellow arrows, which are located three nts upstream of the PAM sequence highlighted in light blue (note: the PAM sequence is not part of the sgRNA sequence). Positions and nt sequences of SmNAGALX1_MISEQ (represented by orange lines) and SmNAGALX2_MISEQ (represented by green lines) primers are shown along the genomic sequence. Amplicon lengths of SmNAGALX1_MISEQ and SmNAGALX2_MISEQ products are underlined in orange and green, respectively.
(TIF)

**S7 Fig. NHEJ-associated indels and representative substitutions observed in lentiviral CRISPR/Cas9 plasmid treated worms by CRISPResso2 analysis.** Multiple sequence alignment (MSA) showing all insertions, deletions, insertions with deletions and representative substitutions (i.e. supported by more than one sequence read) identified in modified sequence reads by CRISPResso2 analysis. Modified sequence reads are aligned with the original unmodified *smnagal* nucleotide (nt) sequence. Alignments are grouped as follows: (A) Experimental treatment groups targeting exon 1 and (B) Experimental treatment groups targeting exon 2. A colour code is used to show the number of sequence reads each indel/substitution appears in

for SmNAGALX1 (red), SmNAGALX2 (dark blue) and dual SmNAGALX1/X2 plasmid treated worms (purple). Nts that are modified by each indel/substitution within the alignment are coloured green. Insertions appear as additional nts not found in the original sequence, deletions appear as lines (-) and substitutions appear as replaced nts located in the same positions as the original sequence. The nt sequences of SmNAGALX1_sgRNA and SmNA-GALX2_sgRNA are highlighted in yellow within the original sequence. DSB sites for SmNAGALX1_sgRNA and SmNAGALX2_sgRNA are indicated by yellow arrows, which are located three nts upstream of the PAM sequence highlighted in light blue (note: the PAM sequence is outside and downstream of the sgRNA sequence). An additional colour code is used to distinguish between start (orange nts) and termination (red nts) codons found in the sequences, which have been highlighted in black.
(TIF)

**S8 Fig. SWAP samples derived from *smp_089290* depleted adult male and female schistosomes show no significant reductions in α-GAL activity.** (A) 6.45 μg of siRNA treated adult male-derived SWAP and (B) 2.44 μg of siRNA treated adult female-derived SWAP were measured for α-GAL using α-GAL colorimetric substrates. Final absorbances were quantified at 410 nm. Using linear trendline equations generated from α-GAL standard curves, α-GAL activity (μg/ml) were calculated for each sample. No statistical significance in α-GAL activity between samples was observed (Student's $t$-test, two tailed, unequal variance).
(TIF)

**S9 Fig. *smnagal* deficiency in adult male and female worms leads to motility defects as assessed by WHO-TDR scoring matrix.** The motility of individual worms (five adult pairs per well) were scored between 4–0 based on the WHO-TDR scoring matrix guidelines; 4 = normal active/paired up, 3 = slowed activity, 2 = minimal activity and occasional movement of head and tail, 1 = absence of motility apart from gut movements and 0 = total absence of motility. The total occurrences of each score was plotted per day for *siLuc* treated and *siSmNAGAL* treated adult female (A) and male (B) worms starting from day two after electroporation up until day seven (day of electroporation considered as day zero). Statistical significance is indicated (General Linear Mixed-Effects Model, NLME and EMMEANS R packages, ** = $p<0.01$). Day one was not included in the statistical analysis due to worms appearing stunned and immobile, likely due to electroporation manipulation. Six wells/biological replicates per siRNA treatment were used for this analysis.
(TIF)

**S10 Fig. Representative fluorescence micrographs of eggs collected from wells of *siSmNAGAL* treated adult female worms reveal a broad spectrum of abnormal morphologies.** Images of eggs from siLuc treated worm pairs depicting (A) vitellocytes under the blue channel (i.e. excitation wavelength = 405 nm and emission wavelength = 461 nm) and (B) auto-fluorescence under the green channel (i.e. excitation wavelength = 488 nm and emission wavelength = 520 nm). Images of eggs from siSmNAGAL treated worm pairs depicting (C) vitellocytes under the blue channel and (D) auto-fluorescence under the green channel. Blue = DAPI+ cells, green = egg auto-fluorescence and white scale bars = 20 μm.
(TIF)

**S11 Fig. *smnagal* deficiency does not affect the presence of vitelline droplets and tyrosinase activity in mature vitellocytes located within adult female worms.** (A) α-NAGAL activity was measured in SWAP derived from si*Luc* treated and si*Smnagal* treated adult female worms (1.08 μg) as described above. Statistical significance is indicated (Student's $t$-test, two tailed, unequal variance, * = $p<0.05$). Representative images of siRNA-treated adult female worms

stained with (B) Fast Blue BB or subjected to (C) diphenol oxidase localisation (20x magnification, red scale bars = 50 μm). Structures labelled include vitellarium (V) and gut (G).
(TIF)

**S1 Video. Movement of *siSmNAGAL* treated worms is substantially impaired when compared to *siLuc* treated worms on day three.** Video footage of (A) siLuc treated and (B) siSmNAGAL treated adult worms was captured using a NexiusZoom stereo microscope (Euromex) and edited with ImageFocus 4 software (Euromex).
(MP4)

**S2 Video. Representative morphologies of eggs derived from siRNA treated female worms.** IMARIS 7.3 software (Bitplane) was used to create a video showing the 360˚ horizontal rotation of a representative egg derived from (A) *siLuc* treated and (B) *siSmNAGAL* treated adult female worms. Blue = DAPI⁺ cells, green = egg auto-fluorescence and white scale bars = 20 μm.
(MP4)

**S1 Table. Initial processing of MiSEQ deep-coverage sequence reads for CRISPResso2 analysis.** The number of sequence reads at each of the three initial processing stages before indel characterisations can be made by further CRISPResso2 analysis is presented. These three initial processing stages are "Reads in inputs" (highlighted in red, first stage), "Reads after pre-processing" (highlighted in blue, second stage) and "Reads aligned" (highlighted in yellow, third stage). "Reads in inputs" refers to the total number of sequence reads from raw MiSEQ sequencing data. "Reads after pre-processing" refers to the number of sequence reads after PCR amplification or trimming artefacts are removed. "Reads aligned" refers to the number of sequence reads that are of high quality (>60% homology to reference amplicon sequence), which are used for indel characterisations. The table also lists the primer pair set and sample (samples amplified by SmNAGALX1_MiSEQ and SmNAGALX2_MiSEQ primers are highlighted in orange and green, respectively) used for each barcoded MiSEQ amplicon library constructed.
(DOCX)

**S2 Table. Detectable frequencies of insertions, deletions, insertions with deletions and substitutions in *smnagal*-edited worms as quantified by CRISPResso2 analysis.** The mutation frequencies attributable to genome editing (i.e. insertions, deletions, insertions with deletions, and substitutions) in smnagal-edited worms as quantified by CRISPResso2 analysis is indicated. The percentage of unmodified sequence reads is included. The primer pair set and sample (samples amplified by SmNAGALX1_MiSEQ and SmNAGALX2_MiSEQ primers are highlighted in orange and green, respectively) used for each barcoded MiSEQ amplicon library constructed are indicated.
(DOCX)

## Acknowledgments

We acknowledge all members of the Hoffmann laboratory and Ms Julie Hirst for assisting in the maintenance of the schistosome lifecycle at Aberystwyth University. We thank Dr Chelsea Davis at the Centre of Excellence for Bovine Tuberculosis laboratory (Aberystwyth University) for assisting in the operation of the Zeiss Axio Imager 2 Microscope. We thank Dr Toby Wilkinson (The University of Edinburgh, UK) for assisting in the statistical analysis of the WormAssayGP2 and adult worm scoring matrix data.

## Author Contributions

**Conceptualization:** Benjamin J. Hulme, Kathrin K. Geyer, Dylan W. Phillips, Wannaporn Ittiprasert, Shannon E. Karinshak, Iain W. Chalmers, Paul J. Brindley, Cornelis H. Hokke, Karl F. Hoffmann.

**Data curation:** Benjamin J. Hulme, Wannaporn Ittiprasert.

**Formal analysis:** Benjamin J. Hulme, Kathrin K. Geyer, Gilda Padalino, Dylan W. Phillips, Wannaporn Ittiprasert, Shannon E. Karinshak, Victoria H. Mann, Iain W. Chalmers, Karl F. Hoffmann.

**Funding acquisition:** Paul J. Brindley, Karl F. Hoffmann.

**Investigation:** Benjamin J. Hulme, Wannaporn Ittiprasert, Shannon E. Karinshak, Victoria H. Mann.

**Methodology:** Benjamin J. Hulme, Kathrin K. Geyer, Josephine E. Forde-Thomas, Gilda Padalino, Dylan W. Phillips, Wannaporn Ittiprasert, Iain W. Chalmers, Paul J. Brindley, Cornelis H. Hokke, Karl F. Hoffmann.

**Project administration:** Benjamin J. Hulme, Kathrin K. Geyer, Josephine E. Forde-Thomas, Wannaporn Ittiprasert, Shannon E. Karinshak, Victoria H. Mann, Iain W. Chalmers, Paul J. Brindley, Karl F. Hoffmann.

**Resources:** Benjamin J. Hulme, Kathrin K. Geyer, Josephine E. Forde-Thomas, Gilda Padalino, Dylan W. Phillips, Wannaporn Ittiprasert, Shannon E. Karinshak, Victoria H. Mann, Iain W. Chalmers, Paul J. Brindley, Cornelis H. Hokke, Karl F. Hoffmann.

**Software:** Gilda Padalino, Dylan W. Phillips, Wannaporn Ittiprasert, Iain W. Chalmers.

**Supervision:** Kathrin K. Geyer, Josephine E. Forde-Thomas, Gilda Padalino, Dylan W. Phillips, Wannaporn Ittiprasert, Shannon E. Karinshak, Victoria H. Mann, Iain W. Chalmers, Paul J. Brindley, Cornelis H. Hokke.

**Validation:** Benjamin J. Hulme, Kathrin K. Geyer, Wannaporn Ittiprasert, Iain W. Chalmers, Paul J. Brindley, Cornelis H. Hokke, Karl F. Hoffmann.

**Visualization:** Benjamin J. Hulme, Kathrin K. Geyer, Dylan W. Phillips.

**Writing – original draft:** Benjamin J. Hulme, Karl F. Hoffmann.

**Writing – review & editing:** Benjamin J. Hulme, Kathrin K. Geyer, Josephine E. Forde-Thomas, Gilda Padalino, Dylan W. Phillips, Wannaporn Ittiprasert, Shannon E. Karinshak, Victoria H. Mann, Iain W. Chalmers, Paul J. Brindley, Cornelis H. Hokke, Karl F. Hoffmann.

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
