## [Decision Letter · Decision Letter 0]

23 Aug 2021

Dear Prof. Hoffmann,

Thank you very much for submitting your manuscript "Schistosoma mansoni α-N-acetylgalactosaminidase (SmNAGAL) regulates coordinated parasite movement and egg production" for consideration at PLOS Pathogens. As with all papers reviewed by the journal, your manuscript was reviewed by members of the editorial board and by several independent reviewers. In light of the reviews (below this email), we would like to invite the resubmission of a significantly-revised version that takes into account the reviewers' comments.

We cannot make any decision about publication until we have seen the revised manuscript and your response to the reviewers' comments. Your revised manuscript is also likely to be sent to reviewers for further evaluation.

Sincerely,

Michael H. Hsieh

Guest Editor

PLOS Pathogens

P'ng Loke

Section Editor

PLOS Pathogens

Kasturi Haldar

Editor-in-Chief

PLOS Pathogens

orcid.org/0000-0001-5065-158X

Michael Malim

Editor-in-Chief

PLOS Pathogens

orcid.org/0000-0002-7699-2064

Reviewer's Responses to Questions

**Part I - Summary**

Reviewer #1: In the manuscript "Schistosoma mansoni α-N-acetylgalactosaminidase (SmNAGAL) regulates coordinated parasite movement and egg production", the authors have characterized an enzymatically active glycosyl hydrolase (SmNAGAL) and have determined that it plays an essential role in the coordination of parasitic movement as well as in the pathways associated with egg production. Targeting SmNAGAL as a drug target could provide a novel approach towards controlling schistosomiasis, which is increasingly showing resistance towards praziquantel. The paper is extremely well written, very detailed and comprehensive. The figures are clear and the legends are self-explanatory. It definitely satisfies all the criteria established by PLoS Pathogens and is extremely beneficial for researchers in this specific field as well as those in related fields.

Reviewer #2: The manuscript of Hulme et al. deals with the molecular and functional characterization of Schistosoma mansoni α-N-acetylgalactosaminidase (SmNAGAL). The authors performed sequence and phylogenetic analyses of putative α-GAL/α-NAGAL proteins showing that Smp_089290 to be the only protein containing functional amino acid residues necessary for α-GAL/α-NAGAL substrate cleavage. Both enzymatic activities were higher in females compared to males, which matched previous analyses. WISH localized SmNAGAL in parenchymal cells, neuronal cells, vitellaria and mature vitellocytes in females. DsRNA-mediated KD in adult worms negatively affected α-NAGAL activity, which correlated with a reduction in motility of adult worms and reduced egg production. This was confirmed by a programmed CRISPR/Cas9 approach editing of SmNAGAL. The authors conclude from their results that inhibition of SmNAGAL may lead to the development of novel anthelmintics.

This is a well-designed and interesting study with a number of new and promising results. However, before this manuscript can be considered for publication, a number of points have to be addressed.

**Part II – Major Issues: Key Experiments Required for Acceptance**

Reviewer #1: No further experiments are required for the manuscript.

Reviewer #2: Lines 378 and 619-620: has this alpha-tubulin isoform (smat1) been tested and proven for its suitability as reference gene for qRT-PCR analyses and the determination of sex-specific differences? In case yes, provide a reference for this (or data, which confirm this), and provide the Smp_number of smat1.

In case smat1 represents Smp_090120, this gene is not equally expressed between males and females, and there is a pairing influence on its transcript level (Lu et al. 2018). This may have influenced the results in Fig. 4, which compared sex-specific transcript patterns. Although this reviewer in principle agrees on the basis of numerous different results also from other groups that SmNAGAL is likely expressed with a female-bias, Smp_090120 is not well suited as a reference gene for this kind of analysis and may have corrupted results and statistics.

Lines 687-690: have the authors checked whether the selected siRNAs against SmNAGAL were specific for this gene and failed targeting also one or more of the other closely related genes? It would be beneficial to show the transcript profiles of the related genes (see line 537 and Smp_334700) to convince the reader the KD was gene-specific for SmNAGAL only.

Lines 906-913: the authors speculate that SmNAGAL may affect vitellocyte development. They assume that the normal development from immature, undifferentiated vitellocytes to mature, fully differentiated vitellocytes is interrupted in knock-down/knock-out females and does not proceed from stage 1 on. Although this seems plausible on the first view, no data have been provided substantiating this idea. The authors may envisage to stain worms (e.g. by lipid staining) to visualize this effect by bright field microscopy (see protocols of the Collins lab) or to perform confocal microscopy to investigate the loss/reduction of mature vitellocytes in KD or KO worms. The authors also speculate about fewer mature vitellocytes containing functionally active tyrosinases in KD or KO worms. In the past the same group has worked in this field and it may be easy for them to substantiate this idea by tyrosinase assays on tissue sections and/or qRT-PCR analyses in KD or KO worms.

The conclusion that inhibition of SmNAGAL may lead to the development of novel anthelmintics is premature at this stage of the functional characterization of this gene.

**Part III – Minor Issues: Editorial and Data Presentation Modifications**

Reviewer #1: Page 6 Line 121 : "Within GH family 27 are the related lysosomal enzymes ..." This sentence needs to be re-written.

Reviewer #2: Line 256: 1,463 bp

Line 409/411: is there a reference for the relaxation protocol? Why is killing with MgCl2 done ahead of fixation in formaldehyde?

Line 537: was this checked also for Smp_334700?

Fig. S2. The color code is not intuitive. Using blue for males may help to get the point faster.

References: italics missing for species names

PLOS authors have the option to publish the peer review history of their article (what does this mean?). If published, this will include your full peer review and any attached files.

Reviewer #1: No

Reviewer #2: No
---

## [Editor Report · Decision Letter 1]

13 Dec 2021

Dear Prof. Hoffmann,

We are pleased to inform you that your manuscript 'Schistosoma mansoni  α-N-acetylgalactosaminidase (SmNAGAL) regulates coordinated parasite movement and egg production' has been provisionally accepted for publication in PLOS Pathogens.

Best regards,

Michael H. Hsieh

Guest Editor

PLOS Pathogens

P'ng Loke

Section Editor

PLOS Pathogens

Kasturi Haldar

Editor-in-Chief

PLOS Pathogens

orcid.org/0000-0001-5065-158X

Michael Malim

Editor-in-Chief

PLOS Pathogens

orcid.org/0000-0002-7699-2064
---

## [Editor Report · Acceptance letter]

10 Jan 2022

Dear Prof. Hoffmann,

We are delighted to inform you that your manuscript, "Schistosoma mansoni  α-N-acetylgalactosaminidase (SmNAGAL) regulates coordinated parasite movement and egg production," has been formally accepted for publication in PLOS Pathogens.

Best regards,

Kasturi Haldar

Editor-in-Chief

PLOS Pathogens

orcid.org/0000-0001-5065-158X

Michael Malim

Editor-in-Chief

PLOS Pathogens

orcid.org/0000-0002-7699-2064